# Bias in downscaled rainfall characteristics

Nicholas J. Potter[1], Francis H.S. Chiew[1], Stephen P. Charles[2], Guobin Fu[2], Hongxing Zheng[1], Lu Zhang[1]

[1]CSIRO Land and Water, Canberra ACT, 2601, Australia
[2]CSIRO Land and Water, Floreat WA, 6148, Australia

*Correspondence to*: Nicholas J. Potter (Nick.Potter@csiro.au)

**Abstract.** Dynamical downscaling of future projections of global climate model outputs can provide useful information about plausible and possible changes to water resource availability, for which there is increasing demand for regional water resource planning processes. By explicitly modelling climate processes within and across global climate model gridcells for a region, dynamical downscaling can provide higher resolution hydroclimate projections, as well as independent (from historical

timeseries) and physically plausible future rainfall timeseries for hydrological modelling applications. However, since rainfall is not typically constrained to observations by these methods, there is often a need for bias correction before use in hydrological modelling. Many bias correction methods (such as scaling, empirical and distributional mapping) have been proposed in the literature, but methods that treat daily amounts only (and not sequencing) can result in residual biases in certain rainfall characteristics, which flow through to biases and problems with subsequently modelled runoff. We apply quantile-quantile

mapping to rainfall dynamically downscaled by NARCliM in the State of Victoria, Australia and examine the effect of this on: (i) biases both before and after bias correction in different rainfall metrics; (ii) change signals in metrics in comparison to the bias; and (iii) the effect of bias correction on wet-wet and dry-dry transition probabilities. After bias correction, persistence of wet states is under-correlated (i.e. more random than observations), and this results in a significant bias (underestimation) of runoff using hydrological models calibrated on historical data. A novel representation of quantile-quantile mapping is

developed based on lag-one transition probabilities of dry and wet states, and we use this to explain residual biases in transition probabilities. This demonstrates that any quantile mapping bias correction method is unable to correct the underestimation of autocorrelation of rainfall sequencing, which suggests that new methods are needed to properly bias correct dynamical downscaling rainfall outputs.

## 1 Introduction

There is a growing and on-going need for information about plausible and possible changes to water resource availability in the future due to climate change. End users of hydroclimate projections need more spatially detailed information as well as information on water metrics for low- and high-flow events, as well as interdecadal metrics (Potter et al., 2018). Dynamical downscaling (such as provided by the NARCliM project for south-eastern Australia, see Sect. 2.1) has potential to provide this type of information, however there remain challenges associated with the use of this data. In this paper, we examine the

suitability of NARCliM projections for providing hydroclimate projections for south-eastern Australia. Specifically, we look at the extent of biases in rainfall, which necessitate daily bias correction, and the effect of quantile-quantile mapping (QQM) bias correction on rainfall sequencing metrics that are important for runoff generation. Subsequent research in a related paper (Charles et al., 2019) focuses attention on the effect of these biases on runoff.

Of particular interest are possible changes to rainfall characteristics that could affect runoff and streamflow. Information on

future changes to rainfall are typically derived from ensembles of global climate models (GCMs), however the spatial resolution of these models is too coarse to provide information at the scale needed for hydrological impact modelling (i.e. catchments or gauges). Downscaling is the process by which finer scale spatial detail is extracted from the larger scale GCM change information (Maraun et al., 2010). Many water resource studies use 'empirical scaling', where historical rainfall observations are scaled (perhaps annually or seasonally) for direct use, or 'statistical downscaling' in which a direct statistical

relationship is developed between rainfall and other atmospheric predictors. These methods are relatively simple to use, and results from empirical scaling typically lie in the middle of the range of results from other downscaling methods (Chiew et al., 2010; Potter et al., 2018). However, as empirical scaling methods rely on the historical record of rainfall, future changes in rainfall sequencing (e.g. changes to multi-day accumulations and wet/dry transitions) and consequent effects on runoff cannot

be properly modelled. Dynamical downscaling, in which a regional climate model (RCM) of finer spatial resolution than the host GCM, generates rainfall sequences independent from historical observations. However, challenges remain with using dynamical downscaling output since rainfall (and other climate variables) are not explicitly constrained by observations (see, e.g., Piani et al., 2010; Chen et al., 2011; Teutschbein and Seibert, 2012). As such, dynamical downscaling outputs typically need to be bias corrected for direct use in hydrological models. In particular, a common feature of dynamical downscaling is

the tendency to underpredict the occurrence of zero- and low-rainfall days, which is sometimes known as the drizzle effect (e.g. Maraun, 2013). RCM output has been bias corrected for applications in Australia in Tasmania (Bennett et al., 2014) and central coast of New South Wales (Lockart et al., 2016) but both these studies found residual biases in multi-day rainfall events, dry spell durations, and autocorrelation of rainfall occurrences. Themeßl et al. (2012) bias corrected RCM output over Europe and found residual biases in rainfall extremes and a modification of the climate change signal.

Water resources in Victoria are shared by urban users, irrigators, industry and the environment. Long-term water strategies and shorter term sustainable water strategies are required for Victoria's water regions by the Victorian Water Act. A key aspect of these water planning processes is accounting for scenarios of climate change as determined by the available science. Cool-season rainfall in Victoria since 2000 has averaged 15% less than the long-term average during the 20th century (Hope et al., 2017). This has been linked to the observed expansion of Hadley Cell circulation (Post et al., 2014). The median scenario of

climate change for Victoria typically has reduced rainfall and runoff later in the century, with slightly larger percentage declines in the western parts of the State (Post et al., 2012; Potter et al., 2016). Providing better information to improve water planning processes includes developing finer spatial resolution projections as well as different metrics of daily rainfall amounts and occurrences.

Most bias-correction methods alter daily amounts with the application of distributional mappings. The temporal structure of

the occurrences is most often unaltered. Further, bias correction can and does affect the magnitude of change signals (Hagemann et al., 2011; Gutjahr and Heinemann, 2013; Dosio, 2016). The underlying assumption of bias correction is that the RCM output faithfully represents the climate processes responsible for rainfall, although the amounts themselves may not be accurate. Water resource projection modelling is concerned with future changes, and so an argument could be made that although the rainfall amounts are biased for hindcast (historical) simulations, they will presumably be equally biased for future

simulations, so that changes can be inferred from comparing biased historical and future rainfall and runoff. However, the sensitivity of runoff to rainfall means that biased rainfall can have large effects on the change signal of runoff (Teng et al., 2015). Furthermore, hydrological models are calibrated to historical rainfall and runoff sequences, and since the distribution of runoff is usually highly skewed, using biased rainfall sequences can distort the distribution of runoff, thus creating large biases in high and low runoff amounts. This makes inferences on the changes to runoff characteristics highly uncertain when

biased rainfall inputs are used.

Bias correction identifies a relationship or mapping between observed historical rainfall and modelled historical rainfall (in this case hindcast RCM rainfall). This mapping when applied to hindcast RCM rainfall results in a distribution of rainfall identical (or very similar, depending on the methods) to the historical observations. This mapping can then be applied to future RCM rainfall, resulting in unbiased future rainfall sequences. Of course, applying the relationship into the future assumes the

bias in RCM rainfall does not change into the future or for different (wetter or drier) climate periods. Bias-correction methods (see Schmidli et al., 2006; Boé et al., 2007; Lenderink et al., 2007; Christensen et al., 2008; Piani et al., 2010; Themeßl et al., 2011; Teng et al., 2015) fall into three main categories:

- Scaling or change-factor methods;

- Non-parametric (empirical) quantile-quantile mapping (QQM); and
- Parametric (distributional) QQM.

Scaling methods simply consider the change in mean and apply a constant factor to correct bias in RCM rainfall. Quantile-quantile mapping matches each quantile (or a selection of quantiles) of the two distributions. This can be done using the

empirical cumulative density or fitting a distribution to both observed and hindcast RCM daily rainfall amounts.

Teng et al. (2015) demonstrated that representing daily rainfall distributions with double-gamma distributions (see also, e.g., Yang et al., 2010) was largely identical to empirical QQM, implying that distributional and empirical approaches give similar results so long as the distribution is sufficiently flexible. Arguably, the choice between non-parametric (empirical) or parametric (distributional) mapping is a representation of the bias-variance trade-off problem. Empirical mapping will reduce

bias to zero, but at the cost of increasing the variance of predictions, since the mapping will be very sensitive to individual amounts. Distributional mapping fits the data across the entire rainfall distribution but can result in the hindcast RCM rainfall not being mapped exactly to the historical distribution. For this study we apply empirical quantile-quantile mapping for each season across integral percentiles as described below. Overall there is a small but relatively unimportant difference between different methods for QQM, at least in the Australian context studied by Teng et al. (2015).

Whereas quantile-quantile mapping can effectively reduce historical error in daily rainfall amounts to zero, albeit with some of the caveats already mentioned, the bias corrected rainfall timeseries could still harbour biases and unrealistic characteristics that will result in runoff biases after being routed through a rainfall-runoff model. Specifically, QQM bias correction cannot remove biases in rainfall sequencing and multi-day accumulations that might not be readily apparent when considering only the daily distribution of rainfall amounts (e.g. Addor and Seibert, 2014; Li et al., 2016; Ines and Hansen, 2006), however

Terink et al. (2010) and Rajczak et al. (2016) contend that bias correction produces reasonable transition probabilities and spell durations.

Unfortunately, it is not easy to tell exactly which characteristics of rainfall drive runoff generation, and in general the sensitivity will depend on catchment physical characteristics, storm type and intensity, as well as antecedent moisture and groundwater stores (Goodrich and Woolhiser, 1991; Bell and Moore, 2000; Beven, 2001). Spectral and multifractal approaches (e.g. Milly

and Wetherald, 2002; Matsoukas et al., 2000; Tessier et al., 1996) show that rainfall variability at shorter timescales is by and large incorporated into soil moisture buffers thus dampening runoff variability at these timescales. However, over timescales of several days and greater, variability in runoff matches variability in rainfall more and more closely. As such, it is evident that large, intense rainfall events (measured perhaps by the upper tail of the rainfall distribution), more seasonal rainfall regimes (Wolock and McCabe, 1999), relatively larger variability of rainfall (Potter and Chiew, 2011), and large multi-day

accumulations of rainfall are most important for runoff generation (Addor and Seibert, 2014), particularly for high flow events (Jaun et al., 2008), and we focus on these kinds of rainfall metrics in this study.

The main aim of the paper is to investigate the effect of bias correction on rainfall characteristics relevant to runoff generation. Given that QQM bias-correction approaches are specifically tailored for correcting daily amounts while retaining the existing sequencing of occurrences, we seek to understand the effect of bias correction on rainfall sequencing and transition

probabilities. Specifically, we investigate whether key rainfall metrics contain biases in dynamically downscaled GCM hindcasts relative to observations. We then examine if bias correction acts to either enhance or moderate any such biases, and whether bias correction affects change signals (i.e. downscaled GCM future relative to downscaled GCM historical). Section 2 describes the data and methods used in this study, Sect. 3 presents results, and Sect. 4 and 5 contain discussion and conclusions.

## 2 Data and Methods

### 2.1 The NARCliM Project

The NSW and ACT Regional Climate Modelling Project (NARCliM; https://climatechange.environment.nsw.gov.au/Climate-projections-for-NSW/About-NARCliM; Evans et al., 2014) is a regional climate change project to deliver high resolution
dynamically downscaled climate change projections. As noted previously, GCMs run at a spatial resolution that is unable to provide meaningful information for decision makers at catchment and basin scale. To resolve subgrid processes, the NARCliM project uses the 'Weather Research and Forecasting' (WRF) regional climate model (specifically the Advanced Research WRF version 3; Evans and McCabe, 2010), which is a mesoscale atmospheric model with many applications both in numerical weather prediction and climate projections (Skamarock and Klemp, 2008), forced by atmospheric variables output from GCMs.
The NARCliM modelling domain is most of south-eastern Australia and the neighbouring Pacific Ocean at 10km × 10km resolution, nested within the larger CORDEX Australasian region (Giorgi et al., 2009) at 50km × 50km resolution (Evans et al, 2014). WRF reads in output from a GCM along its lateral and lower boundaries and simulates the climate on a finer resolution within those boundaries (Skamarock et al., 2008). Out of the 23 CMIP3 (Meehl et al., 2007) GCMs available at the time the NARCliM project started, four were eventually chosen for downscaling: MIROC3.2 (medres), ECHAM5, CCCM3.1
and CSIRO-Mk3.0 (Evans et al., 2013). Initially, GCMs that did not adequately represent climate dynamics of the region were eliminated. Based on a meta-analysis, GCMs were then ranked, and independence of model error, as well as future changes, was assessed to represent the range of model ensemble (Evans and Ji, 2012a). A similar procedure was applied to select the configurations of WRF used to downscale each host GCM (Evans and Ji, 2012b; see also Ji et al., 2016; Olson et al., 2016). WRF models were evaluated against 2-week heavy rainfall events with an intent to select the best possible RCM configurations
for rainfall generation while also accounting for model uncertainty (Olson et al., 2016).
Overall, there is reasonable confidence in NARCliM projections generally, for both rainfall and temperatures (Evans et al., 2012; Olson et al., 2016; Ji et al., 2016), particularly at daily scale for rainfall (Gilmore et al., 2016), although NARCliM has a quantitative cold and wet bias generally (Ji et al., 2016). The underlying performance of the RCM component of WRF has been tested extensively (Evans et al., 2012; Andrys et al., 2016; Ekström, 2016; Olson et al., 2016; Ji et al., 2016; Gilmore et
al., 2016). The three WRF physics configurations are labelled R1, R2 and R3 (see Evans et al., 2014, Table 1), detailed information is provided elsewhere (Evans et al., 2012; Ekström, 2016; Gilmore et al., 2016; Ji et al., 2016). The full NARCliM model ensemble thus consists of twelve members (four selected GCMs, each downscaled by the three RCM configurations). Ji et al. (2016) evaluated model output against AWAP (see below) rainfall, demonstrating good representation of precipitation processes at a wide range of timescales, but conclude that bias correction is still needed before applying the data. For
hydrological applications, the R2 configuration is recommended (Olson et al., 2016). Although we consider the entire modelling ensemble in this paper, in a related paper (Charles et al., 2019) we use only that specific RCM physics scheme for modelling runoff. We discuss the physical credibility of the NARCliM ensemble in greater detail in Sect. 4 below.

### 2.2 Daily Rainfall Data

Daily accumulated precipitation from the NARCliM 12-member WRF-downscaled ensemble (referred to from here onwards
as 'NARCliM rainfall') is produced at approximately 10km × 10km grids, corresponding to the NARCliM domain covering south-eastern Australia (Evans et al., 2014; Ji et al., 2016), which was bilinearly interpolated to a regular grid using Climate Data Operators (CDO; https://code.mpimet.mpg.de/projects/cdo). NARCliM provides bias corrected data (Evans and Argüeso, 2014), which applies a parametric gamma distribution quantile-quantile mapping procedure, which is similar in many respects to the non-parametric procedure we apply to the raw NARCliM data in this paper (Teng et al., 2015).
The historical (baseline) period for NARCliM projections is 1990–2009, and we relate NARCliM historical rainfall features to observations and NCEP/NCAR reanalysis over the same period in this paper. Realisations of future rainfall follow the A2

Scenario of the Special Report on Emissions Scenarios (SRES) (Nakićenović et al., 2000) at 2060–2079. Change signals presented later are thus averages over 2060–2079 compared to 1990–2009. Observed rainfall data is obtained from the Australian Water Availability Project (AWAP; Jones et al., 2009). This is a 0.05°×0.05° gridded dataset interpolating observations from point rainfall records from the Australian Bureau of Meteorology. The AWAP rainfall observations are projected over a regular latitude-longitude grid, hence the need for interpolation of NARCliM data to be aligned with this. The AWAP rainfall data is then regridded by using weighted averages of AWAP gridcells overlapping the 0.1°×0.1° AWAP grid.

## 2.3 Quantile-Quantile Mapping Bias Correction

Quantile-quantile mapping bias correction works by estimating the cumulative density function for observed and modelled historical daily rainfall amounts: $F_o$ and $F_m$. These are then combined to produce a mapping function:

$$P_o = F_o^{-1} \circ F_m \ (P_m)$$

(Fig. 1). The map $F_o^{-1} \circ F_m$ thus returns the observed daily distribution (or approximately so, depending on the method) when applied to the modelled historical timeseries, and a bias corrected future timeseries when applied to the modelled future timeseries. We use the R package 'qmap' (Gudmundsson et al., 2012), to estimate the cumulative density functions for the mapping function. This estimates quantiles for both observed and modelled non-zero rainfall at integral percentiles including 0 (minimum) and 100 (maximum). The quantile for a particular daily rainfall amount is then estimated using linear interpolation between percentiles, and linear extrapolation in case of future modelled rainfall lying outside the historical distribution. Compared to using the empirical distributions directly, differences between the bias-corrected modelled rainfall distribution and the distribution of observations can occur (of the order of 2–3%) because the interpolation between large rainfall percentiles (particularly 99 to 100) will not match observed percentiles exactly. As noted by Teng et al. (2015), sufficiently flexible approaches to bias correction give very similar results. QQM bias correction in this way was applied separately to each three-month season (i.e. DJF, MAM, JJA and SON) in each grid cell independently, using the full historical period (1990–2009) as calibration period, to both historical (1990–2009) and future (2060–2079) periods.

## 2.4 Defining Transition Probabilites

Whereas QQM bias correction can correct the daily distribution exactly, daily bias correction is not set up to correct sequences and accumulations (Addor and Seibert, 2014). To this end, we consider not only how bias correction affects daily metrics of rainfall, but also the sequencing of wet days that produce runoff. One way of measuring this is through transition probabilities of wet and dry sequences. To this end, we consider a simple two-state Markov Chain rainfall occurrence model. Here, the probability of a wet or dry day depends on whether the previous day was wet or dry. A 'dry' day can be defined as either zero rainfall or rainfall below a given threshold (such as 1 mm). Define the wet-to-wet transition probability (i.e. the probability of a wet day following a wet day) as $w = \Pr(W_n|W_{n-1})$ and the corresponding dry-to-dry transition probability as $d = \Pr(D_n|D_{n-1})$. These determine the Markov Chain since $\Pr(D_n|W_{n-1}) = 1 - w$ and $\Pr(W_n|D_{n-1}) = 1 - d$. Further, the probability of the occurrence of a dry day, $p$, is fully determined by the $w$ and $d$ parameters, as given by Cox and Miller (1965):

$$p = \frac{1 - w}{2 - d - w}$$

Equivalently,

$$w = 1 - p\frac{1 - d}{1 - p}$$

This relationship can be plotted for a range of probabilities $p$ (dotted lines in Fig. 10). Note that if a series of occurrences of dry and wet days has zero autocorrelation (i.e. the state probability is independent of the rainfall state in the previous day), then it follows that $\Pr(D_n|D_{n-1}) = \Pr(D_n|W_{n-1}) = \Pr(D_n) = p$. As such, the diagonal line where $p = d$ (dashed line in Fig.

10) corresponds to an independent (i.e. zero autocorrelation) series of occurrences. The area above and to the right of the dashed line corresponds to a series of occurrences with positive autocorrelation (i.e. $\Pr(D_n | D_{n-1}) > p$, so that dry sequences are more likely to persist), whereas the area below and to the left of the dashed line corresponds to series with negative autocorrelation. The framework developed above is used in Sect. 3.3 as a novel way to represent the relationships between state-transition probabilities and rainfall quantiles to investigate the effect of bias correction on transition probabilities.

## 3 Results

### 3.1 Assessment of Regional Performance of Modelled Rainfall

Figure 2 shows mean annual rainfall across Victoria from (a) AWAP observations, (b) downscaled (NCEP/NCAR) reanalysis and downscaled-GCM (c) historical and (d) future. ECHAM5-R1 was chosen as a representative GCM-RCM ensemble member here as it had the median historical regional rainfall across Victoria from the NARCliM model ensemble. The spatial rainfall fields downscaled from both the reanalysis and GCM both show reasonable agreement with the spatial pattern of the observed mean annual rainfall, with relatively larger rainfall across the mountain ranges to the east of the State and across the southern coast, and less rainfall to the more arid north-west region. However, both rainfall fields are evidently positively biased with around 100–200 mm of excess rainfall consistently across most of the State (Fig. 2f, g), relatively more for the far eastern part of Victoria for the reanalysis data and across the mountain ranges for the GCM data, interspersed with small patches of negative bias (observations larger than downscaled data). Averaged across Victoria, the mean absolute bias is approximately 26% for the downscaled reanalysis and 43% for the downscaled GCM data. In comparison, the absolute change signal averages 3%, rising to around 10% in the eastern part of Victoria (Fig. 2h).

### 3.2 Bias Correction of NARCliM

Consistent with Fig. 2, the raw NARCliM rainfall is mostly wetter (positive bias) across Victoria (Fig. 3a), except for a tendency towards underprediction in the south-east coast for some models. The quantile-quantile bias correction method is formulated to correct historical quantiles of rainfall exactly (when applied to the same time period used for calibration) so that bias-corrected mean annual rainfall, as well as any quantiles, are approximately equal (Fig. 3b). However, the method used here does not correct high rainfall quantiles (e.g. P99 and above) exactly, due to the interpolation between quantiles as described in Sect. 2.3 (Fig. 4b). This residual bias appears to be randomly distributed spatially and results in bias-corrected mean annual rainfall not being exactly corrected. However, this effect is generally less than 5% of the mean annual rainfall. This effect can be removed entirely by using empirical density functions rather than interpolated values, but with the effect of increasing prediction uncertainty.

Figure 5 shows the distribution of bias for different rainfall metrics before bias correction and the residual bias after QQM bias correction. The bias in raw NARCliM ranges from 5% to 50% for all percentiles (Fig. 5a), increasing as the percentile increases (i.e. more relative bias for higher rainfall amounts). As with Fig. 3, bias at all percentiles is effectively reduced to zero after bias correction, although the residual bias is relatively larger at larger percentiles (higher rainfall), similarly to Fig. 4b. NARCliM rainfall is overestimated at all seasons and months before bias correction (Fig. 5b), although winter rainfall is relatively less biased than summer rainfall. Bias correction reduces bias to zero annually and seasonally, since QQM is applied to each season separately. Since the intra-seasonal relative monthly rainfall amounts are not exactly equal to the observed amounts, seasonal bias correction occasionally overcorrects bias, particularly in February, April, May and June, with the bias corrected rainfall in these months being less than observed whereas they were overpredicted before bias correction; this could be reduced to zero using monthly correction factors, although at the potential cost of overfitting. Overall though, the absolute relative bias is reduced and closer to zero in all months compared to the raw RCM monthly amounts.

Figure 5c shows relative bias in rainfall sequencing related metrics. Autocorrelation of rainfall amounts is underpredicted before bias correction, and the magnitude of this bias actually increases after bias correction. Whereas QQM reduces bias in dry-dry transition probabilities (i.e. the probability of dry sequences persisting), bias in wet-wet transition probabilities increases after bias correction so that, similarly to autocorrelation, the probability of wet spells persisting is considerably underpredicted after bias correction. We examine this in more detail in Sect. 3.3 using the transition probability framework developed in Sect. 2.4. Since dry spells are directly related to dry-dry transition probabilities, the bias in mean and maximum dry spells is well corrected, whereas maximum 3-day rainfall accumulation and wet-spell occurrences all have negative bias after QQM bias correction. Different percentiles of 3-day rainfall accumulation (calculated as percentiles of a 3-day moving sum of rainfall timeseries) have different residual bias (Fig. 6). Three-day accumulation percentiles below the 80th are all slightly overestimated after bias correction, but above the 80th percentile, a large residual underestimation is present. This reduces at around the 99th percentile but is moderated somewhat for the 3-day maximum (i.e. 100th percentile). As noted in section 2.1, WRF models were selected according to their skill in reproducing selected 2-week periods of heavy rainfall. This provides a potential explanation for the smaller bias in 3-day maxima relative to 3-day 99% rainfall.

It is likely that underpredicting wet spell occurrences and persistence (Fig. 5c and Fig. 6) will result in runoff from the bias corrected rainfall being underpredicted too. To explore this, runoff was modelled from bias corrected rainfall and observed potential evapotranspiration data using GR4J (see Charles et al., 2019 for more details). Figure 7 plots the percentage difference in bias-corrected ensemble-median mean annual runoff for each 0.1°×0.1° cell compared to mean annual runoff modelled using AWAP observed rainfall. The ensemble median of runoff across Victoria is underpredicted by between 10–20% across almost all of Victoria suggesting that the residual bias in wet spell occurrences and persistence is problematic for runoff modelling. Whereas the smallest percentage biases appear to be over the high-runoff producing region (Fig. 7c), this region has the highest absolute biases with bias in runoff of more than −20 mm (Fig. 7b). Characteristics and biases of runoff from bias corrected NARCliM rainfall is explored in more detail by Charles et al. (2019).

## 3.3 Residual Bias in Rainfall State Transition Probabilities

The results in this study demonstrate that dry-dry transition probabilities for NARCliM have low residual bias (possibly due to the emphasis in QQM on preserving zero-rain occurrences), but that wet-wet transition probabilities have more bias (i.e. they are closer to zero, see Fig. 5c) after QQM bias correction. This results in the persistence of wet spells being underestimated even though the volumetric amount of rainfall is, by design of QQM bias correction, equal to observed rainfall at any gridpoint. After bias correction, dry-dry transition probabilities for a 1 mm threshold are reduced, but still have a small negative bias (Fig. 5c). Figure 8 shows the dry-dry transition probabilities calculated from observed rainfall (top left), bias corrected WRF-downscaled reanalysis (middle column), bias corrected historical WRF-downscaled GCM (right column). Both the bias corrected WRF-downscaled reanalysis and bias corrected WRF-downscaled GCM results show a spatial pattern very similar to observations with higher dry-dry transition probabilities to the northwest of the State, and at similar places along the southern coastline. However, the reanalysis and more so the GCM result has a lower dry-dry transition probability across almost all of the region (Fig. 8). As such, dry spells from the bias-corrected model output are likely to be shorter in duration and less common than that from the observed rainfall (although bias correction does reduce the bias in dry spells somewhat compared to the bias in the raw data as seen in Fig. 5c).

Whereas the dry-dry transition probabilities were largest in the north-west, drier, part of Victoria, the wet-wet transition probabilities are largest over the high-runoff producing region (Fig. 9), which corresponds to the high-relief, high-altitude part of the State. As with the dry-dry transition probabilities, both bias-corrected downscaled reanalysis and bias-corrected downscaled GCMs reproduce the spatial pattern of wet-wet transition probabilities, but there is considerable residual bias in these probabilities across the entire region. The residual bias in WRF-downscaled GCM transition probabilities is over 10% over most of Victoria. This results in underestimation of wet spell occurrences and durations and multiday accumulations of

rainfall (Fig. 5c). The bias in wet-wet transition probabilities is more problematic for modelling runoff than the bias in dry-dry transition probabilities, not only because it is of larger magnitude, but because:

- runoff is sensitive to multiday wet spells
- the larger wet-wet probabilities occur in high runoff producing areas, which we would like to model correctly for regional water availability projections
- QQM bias correction reduces the bias in dry-dry transition probabilities but increases the magnitude of the bias in wet-wet transition probabilities (Fig. 5c).

Figure 10 shows the observed (green), raw (blue) and bias-corrected (red) historical downscaled GCM rainfall amounts for a sample grid cell overlaid on the transition-probability space developed in Sect. 2.4. Other grid cells and GCMs show very similar responses (as can be seen in the low spread of results for $d$ and $w$ in Fig. 5c). Quantile-quantile mapping bias correction equates the quantiles for each rainfall amount such that equal rainfall amounts for observations and bias-corrected rainfall occur on the same probability contours (dotted lines in Fig. 10), with raw values translated along the rainfall amount curve. That is, values on the blue line in Fig. 10 map to corresponding values on the red line; the slight variation between the lines is due to different bias corrections in each season. As such, wet-wet and dry-dry transition probabilities for a given rainfall threshold (e.g. 1 mm) for bias-corrected rainfall are equal to the transition probabilities for the corresponding amount in the raw data. For example, in Fig. 10, the exceedance probability for 1 mm in the observed data (green line) is 0.774. The corresponding quantile in the raw data (blue line) is 2.675 mm. This amount is mapped to 1 mm in the bias-corrected data (red line), and the corresponding wet-wet and dry-dry transition probabilities for 2.675 mm are identical to the transition probabilities for 1 mm in the bias-corrected data. Recall that points above and to the right in Fig. 10 correspond more (positive) serial correlation. This implies that the observed rainfall timeseries contains more correlation structure in the sequence of wet and dry spell occurrences than the modelled rainfall sequence, and that QQM bias correction cannot rectify this since daily QQM retains the autocorrelation structure of the raw time series since daily amounts are simply rescaled. We surmise that, as bias correction is ultimately intended to produce physically plausible rainfall, bias correction methods that adjust occurrences are needed to properly correct biases for hydrological modelling.

**3.4 Change signals**

Here we examine change signals in rainfall metrics (i.e. percentage difference in RCM future relative to RCM historical averages) specifically looking at whether bias correction alters the change signals. Figure 11a shows the change signals in different rainfall percentiles. For the raw data, there is a small decrease in low to moderate rainfall amounts less than the non-zero 40th percentile and a future increase in non-zero percentiles above 50%. The nature of QQM bias correction means that raw and bias corrected equal percentiles cannot be meaningfully compared. Nevertheless, a similar pattern is found with the bias corrected data, namely that larger rainfall amounts have larger relative changes than smaller rainfall amounts.

Figure 11b shows change signals in mean annual, seasonal and monthly average rainfall. The magnitude of the median change signal in mean annual rainfall is around –5%, and seasonal changes are comparable to the annual change except for SON rain which has a decrease projected by the NARCliM ensemble of around 20% (found to be statistically significant by Olson et al., 2016). Compared to the raw bias in mean and seasonal rainfall (Fig. 5b) of between 25%–50%, these change signals are between one-half and one-tenth of the bias. After bias correction, there is little difference in the magnitude and direction of change in seasonal and monthly averages. However, the mean annual rainfall change is moderated somewhat, and this is problematic since mean annual changes are most often considered in regional projection applications (a discussion on bias correction effects on change signals is considered in the next section, and Charles et al. (2019) discuss this in the context of the present study in more detail).

Although the residual bias in rainfall sequencing metrics is not eliminated, and in some cases (e.g. wet-wet transition probabilities) is actually increased, after bias correction (Fig. 5c), Fig. 11c shows that the change signals in rainfall sequencing

metrics is largely unaffected by bias correction. With the exception of maximum 3-day rainfall accumulation, the distributions of sequencing change signals are largely identical before and after bias correction. The difference in change signal for maximum 3-day rainfall accumulation is presumably related to the relatively larger increase in large rainfall events (e.g. P99 in Fig. 11a).

## 4 Discussion

We demonstrate in Sect. 3.4 that change signals (future mean relative to historical) in rainfall metrics can be considerably smaller than the bias (modelled historical relative to observed historical). On the one hand, this seems problematic since biases in processes can be considered so large that the changes are insignificant. On the other hand, there is no particular legitimacy for this viewpoint, certainly not from a statistical sense. The magnitude of bias does not provide any sort of confidence level in changes to rainfall metrics. However, given such relatively large biases, it is reasonable to assume that there are some errors in the way particular climate processes are modelled, either through the host GCM or the RCM. It would be desirable to understand the reasons and climatic process responsible for biases and assess whether these processes are unrealistic, as well as whether these biases render the changes physically implausible.

We acknowledge that the underlying performance of GCMs in accurately simulating climate dynamics of the region under consideration is extremely important. In the context of the study area in the current manuscript, South-Eastern Australia, the selection of GCMs representing important climate processes has been an on-going research strand both for the CMIP3 climate model ensemble (Smith and Chandler, 2010; Kirono and Kent, 2011; Kent et al., 2013; CSIRO, 2012; Evans and Ji, 2012a; McMahon et al., 2015), which is used for the current NARCliM dataset, as well as the CMIP5 ensemble (see CSIRO and Bureau of Meteorology, 2015; Hope et al., 2016). As mentioned in the introduction, the RCM component of NARCliM, WRF, has been tested extensively, including the general cold and wet bias over south-eastern Australia. It is suggested that the wet bias (Fig. 3a) is related to subgrid cloud cover representations (see, e.g., García-Díez et al., 2015; Di Virgilio et al., 2019) and correction of this is the subject of current, ongoing research (Di Virgilio et al., 2019). Current work (NARCliM1.5) is continuing to develop the modelling framework at a higher spatial resolution (5km × 5km), using CMIP5 models and improved RCM configuration (Downes et al., 2019).

The use of models that produce plausible climate dynamics is of course desirable, however in practice it is not necessarily always possible. Apart from the fact that the 'best' models identified by the above references differ according to the criteria used, using a dynamical downscaling ensemble for hydrological applications is an opportunistic endeavour, relying largely on existing data products, which have been prepared with many applications in mind, not just hydrological applications. As such, it is not always practical to choose the GCMs and RCMs that best represent climate dynamics important for hydrological applications; many studies have also contended that accuracy in representing historical conditions is no guarantee that future changes are correctly modelled (e.g. Knutti et al., 2010; Racherla et al, 2012).

As mentioned in the introduction. Evans and McCabe (2010) examined the RCM component (WRF) of NARCliM, concluding that the El Niño-Southern Oscillation, the chief climate process modulating interannual variability of rainfall (Power et al., 1999), was well modelled over south-eastern Australia. Evans and McCabe (2010) also concluded that the severity and duration of recent prolonged droughts over south-eastern Australia were also captured, although the spatial pattern was not characterised exactly. The sub-tropical ridge, which determines the seasonal positioning of storm tracks over southern Australia, was less well represented by WRF (Andrys et al., 2016). Based on the results cited here, we have confidence in the modelling setup of NARCliM to represent atmospheric circulation for southern Australia reasonably well, although we acknowledge that bias correction of NARCliM for end-user applications should consider model skill in atmospheric circulation.

In general, bias correction does not tend to alter the change signals in rainfall metrics (with the exception of 3-day accumulation and low rainfall percentiles). Nevertheless, small differences in rainfall metrics can result in large differences in runoff metrics

and other water availability measures (e.g. low flows and high flows). High runoff and even average runoff amounts can be very sensitive to 3-day rainfall accumulation, which we saw can be altered through daily bias correction. It is recommended that the effects of bias correction are included in any uncertainty analysis undertaken.

Section 3.4 of the manuscript shows that bias correction can affect change signals (future relative to historical) of different hydroclimatic metrics (see also Hagemann et al., 2011; Gutjahr and Heinemann, 2013; Dosio, 2016). Under the assumption that bias is time invariant, Gobiet et al. (2015) argue that bias correction improves the accuracy of climate change signals. Cannon et al. (2015), however, argue that trend-preserving methods should be used (see also Li et al., 2010; Wang and Chen, 2014). Maraun (2016) and Maraun et al. (2017) summarise the debate surrounding the use of trend-preservation methods and conclude that the decision should be informed by the credibility or otherwise of the GCMs in representing the processes driving the changes. This further highlights the need for informed selection and screening of GCMs at the start of the modelling process. However, we argue that there is value in reporting both pre- and post-bias correction future changes in light of the difficulties involved in model selection and assessment, particularly in the case of pre-existing and computationally expensive projections such as dynamically downscaled ensemble such as NARCliM.

The simple QQM method used here does not consider spatial correlation between rainfall gauges or gridcells at all. Maintaining spatial correlations is clearly important for runoff generation and neglecting this can lead to 'inflation'. Inflation refers to a phenomenon in bias correction (Maraun, 2013) or statistical downscaling (von Storch, 1999) where an unmeasured predictand variable is estimated using the predicted values from a statistical model. Since models contain error, the variance of a timeseries of predicted values is expected to be less than the variance of the true time series of the variable. In the present context of bias correcting rainfall from RCMs, Maraun (2013) demonstrates that bias correction reduces subgrid spatial heterogeneity compared to actual precipitation, and that this is particularly problematic when GCM or RCM resolution is much greater than that of observations. In this case, the spatial correlation between gauges is increased. As a result, large rainfall amounts become overestimated and low amounts underestimated. Preserving the correct spatial correlation between gauges or gridpoints is an important issue, and the issue of unintended spatial effects of (temporal) bias correction is compounded by applying bias correction independently to each gridcell, as we have done here – although this tends to reduce subgrid spatial correlation (see Hnilica et al., 2017). Maraun (2013) recommends aggregating catchment rainfall prior to bias correction to reduce the issue of inflation, and Charles et al. (2019) examine this in more detail in relation to catchment runoff production. Variance inflation due to differing grid cell sizes (Maruan, 2013) is less an issue for the current study, as NARCliM grid cell size is comparable to that of the gridded rainfall observations, and the next generation of dynamically downscaled climate projections (Downes et al., 2019) is to be provided at 0.05°×0.05° resolution, identical with the gridded rainfall products used in Australia. However, the issue of using a bias correction methodology that corrects daily amounts (and more generally temporal structure) while preserving spatial structure across catchments and basins remains a challenge and is a direction for further research. Equally, Bárdossy and Pegram (2012) noted that RCM rainfall was considerably less spatially correlated than observations in the Rhine basin, even after QQM bias correction. Lower modelled spatial correlation in rainfall would lead to underestimated extreme flow events (even at the level of 1-year return period events), with a consequent underestimation in areally averaged inflows. Bárdossy and Pegram (2012) recommend pre-correcting spatial correlation using matrix recorrelation methods or by using a sequential recorrelation techniques, and this should be taken into consideration when applying bias correction for projection applications.

Although in this study, rainfall alone is bias corrected; for runoff applications, temperature or a suitable representation of potential evapotranspiration is needed. Methods exist for correcting rainfall and temperature simultaneously (e.g. Hoffmann and Rath, 2012; Piani and Haerter, 2012; Mehrotra et al., 2018). However, potential evapotranspiration has a second-order effect on runoff compared to rainfall (Chiew, 2006; Potter et al., 2011), and bias correction was shown to not significantly affect the inherited relationships between rainfall and temperature (Wilcke et al., 2013). Certainly, the host GCM and RCM

should correctly represent relationships between atmospheric variables in the study region, further highlighting the need for climate model assessment in construction of the model ensemble.

Another important consideration is the relevant metrics to be considered by end users. Bias correction by season, for example, can alter change signals annually, and care must be taken as to which metrics are of interest, and which are the most appropriate bias correction methods to apply in order to properly account for the metrics of interest. Certainly, caution must be applied when considering rainfall and runoff metrics that were not considered when applying bias correction to projections. Low flow metrics are particularly problematic (Potter et al., 2018), where different downscaling and bias correction methods can give very different answers.

Although daily bias correction methods as outlined in this paper tend to result in residual bias in multi-day metrics, generally change signals in transition probabilities are very similar before and after bias correction. This information could thus potentially be extracted from RCMs to drive local weather generation or stochastic methods to provide future rainfall projections that can be suitable for local hydrological projections. Maintaining interannual and multi-decadal correlations, as well as spatial correlations between rainfall gauges, remains a challenge for stochastic methods, however.

**5 Conclusions**

Projections of future changes to rainfall and runoff from dynamically downscaled climate models often necessitates a form of bias correction to rainfall fields to obtain sufficiently realistic rainfall inputs for hydrological models. Dynamical downscaling offers potential benefits to regional hydroclimate projections, such as the ability to better model daily rainfall metrics, low flow metrics (after modelling runoff with a hydrological model), and finer spatial scale information, but comes with challenges related to bias. Whereas bias in rainfall amounts can be corrected using quantile-quantile mapping (QQM) methods, biases in rainfall occurrences (such as rainfall autocorrelation, dry-dry and wet-wet transition probabilities) are retained and in some cases increased with QQM.

The relative magnitude of change signals (future RCM to historical RCM) of the different rainfall metrics examined here is typically less than the magnitude of the bias. Mean annual rainfall change is an order of magnitude smaller than the bias in mean annual rainfall but seasonal changes are closer to half of the bias in seasonal averages. Although this might call into question the validity of the change signal, one approach is to assume that the magnitudes of the changes are responsive to changing greenhouse gas emissions, insofar as the changing atmospheric processes are realistically modelled by the RCM. Indeed, this is the basic premise behind empirical scaling, i.e. that the change is the authentic signature of the climate modelling especially since the RCMs are not explicitly tuned to observed rainfall.

Individual percentiles and seasonal totals are, by design, effectively reduced to zero using QQM. Some interpolation and extrapolation occurs in the approach used here, so there is some random residual bias in higher percentiles (i.e. high rainfall amounts). This can be eliminated altogether by using the exact empirical density functions, but at the cost of increased predictive uncertainty. Using empirical densities also raises problems with extrapolation past historical amounts. Monthly totals retain some residual bias because of compensating biases within each season due to small errors in rainfall seasonality by the RCMs. Metrics associated with rainfall sequencing (e.g. serial correlation, wet-wet and dry-dry state transition probabilities and quantiles of 3-day accumulation) all have significant residual bias, particularly so for wet-wet state transition probabilities in which the magnitude of bias in raw RCM historical runs is amplified after bias correction. This leads to a considerable underestimation of mean annual runoff after rainfall is routed through a hydrological model because runoff is very sensitive to multiday accumulations of rainfall and sequencing of wet spells in particular.

An analysis of the lag-one transition probabilities (i.e. wet state to wet state and dry state to dry state) showed that NARCliM rainfall had transitions to different states that are more random (i.e. more independent) compared to observed rainfall. QQM bias correction is unable to correct these transition probabilities as QQM retains the transition probabilities for any particular

quantile. Since persistence of wet spells is critical for runoff generation, a different approach to bias correction is needed to successfully use NARCliM for runoff projections that can correct rainfall sequencing to better represent the observed correlation structure in wet and dry occurrences.

Change signals in annual, seasonal and monthly average rainfall as well as rainfall sequencing metrics are largely preserved after bias correction, with the exception of maximum 3-day rainfall accumulation. However, there is a slight tendency for DJF and MAM rainfall change signals to increase after bias correction and this leads to a tangible reduction in the magnitude of the projected decrease in mean annual rainfall. This is problematic for applications since mean annual change is the most commonly used metric for hydroclimate projections. One possible solution is to rescale the bias corrected rainfall according to raw changes signals but this depends on whether we believe the raw or the bias corrected change signal is correct. However, the fact that rainfall sequencing metrics (such as state transition probabilities and daily rainfall autocorrelation) are largely unchanged by bias correction suggests the possibility of using this information to drive either weather-generation models or stochastic/resampling-based bias correction methods to produce hydrologically realistic rainfall sequences for hydroclimate projection applications.

### Acknowledgments

This study was funded by the Victorian Department of Environment, Land, Water and Planning (DELWP) through the Victorian Water and Climate Initiative. The authors express thanks to the NARCliM Project (https://climatechange.environment.nsw.gov.au/Climate-projections-for-NSW/About-NARCliM) for freely providing the climate data. We thank three anonymous referees for their helpful and constructive comments and feedback.

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

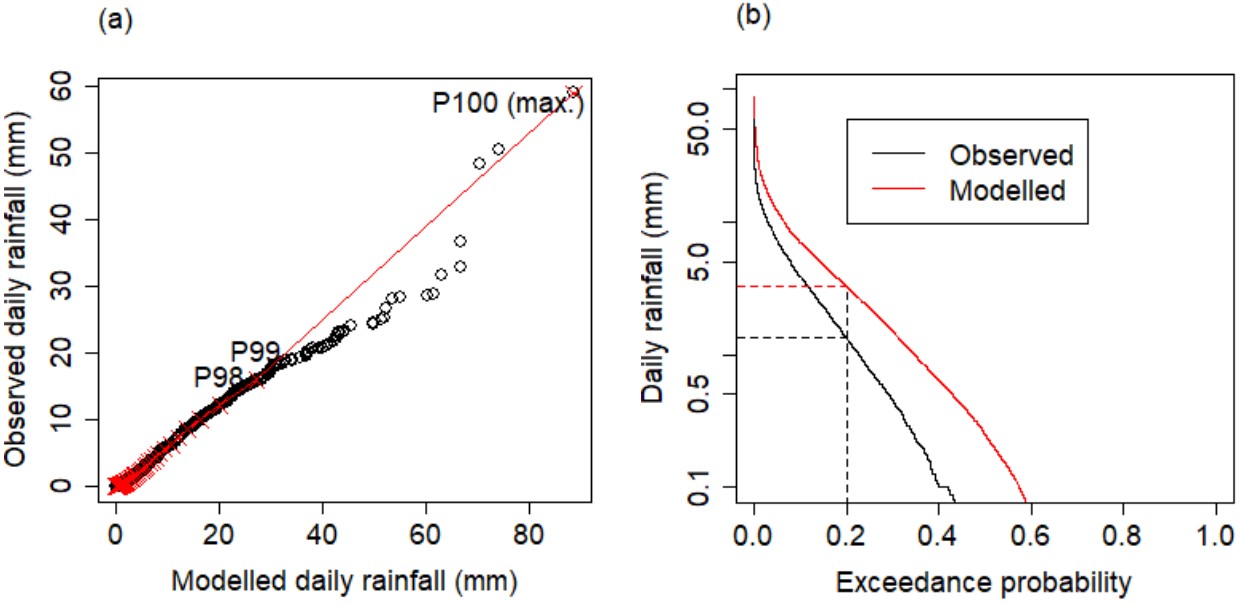

20  **Figure 1: Schematic of QQM bias correction. (a) Empirical cumulative density functions for both observed and modelled rainfall in a given gridcell. Percentiles are estimated from both distributions, which are equated in bias correction to generate a mapping function (b). Values lying between percentiles or outside the modelled maximum value are interpolated or extrapolated linearly.**

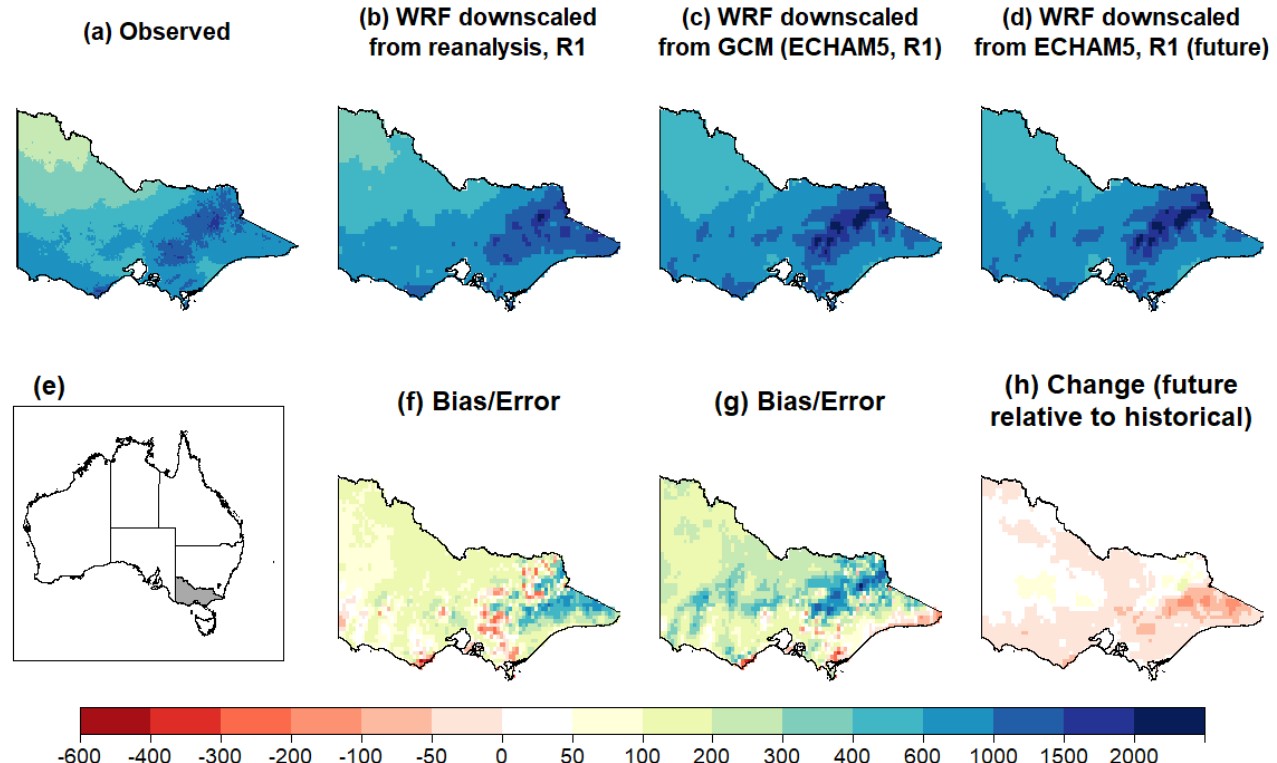

**Figure 2: Regional mean annual rainfall (mm yr⁻¹). Inset map (e) shows the location of the State of Victoria in Australia. Other panels show: (a) observed (AWAP) rainfall; (b) rainfall downscaled from reanalysis (NCEP/NCAR); (c) historical rainfall downscaled from median GCM (ECHAM5/R1); (d) future rainfall downscaled from GCM; (f) bias in reanalysis downscaling (compared to observed); (g) bias in GCM downscaling; (h) change factor of GCM downscaled rainfall.**

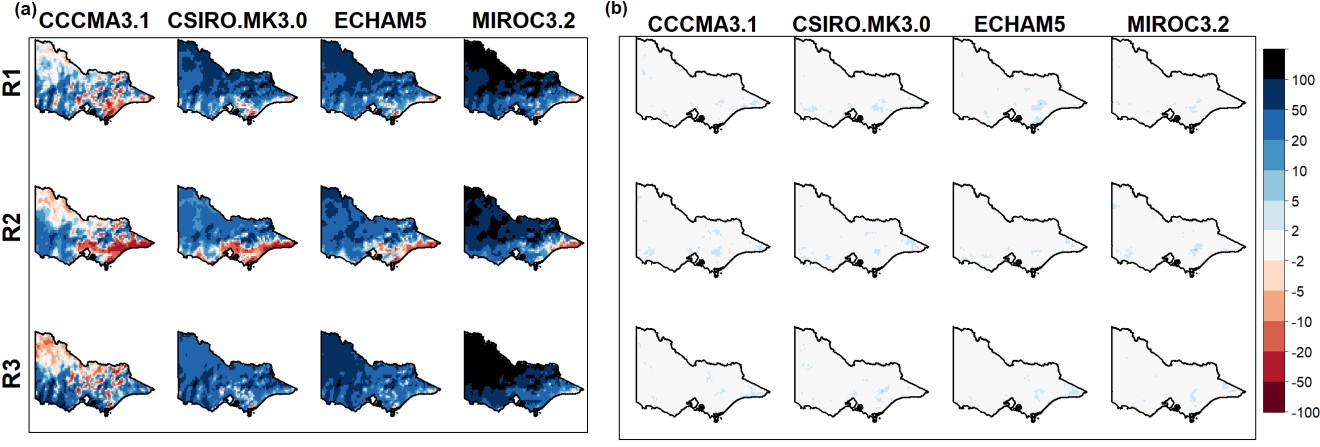

**Figure 3: Percentage bias in WRF-downscaled mean annual rainfall from the GCM/RCMs indicated: (a) raw data; (b) residual bias (after bias correction).**

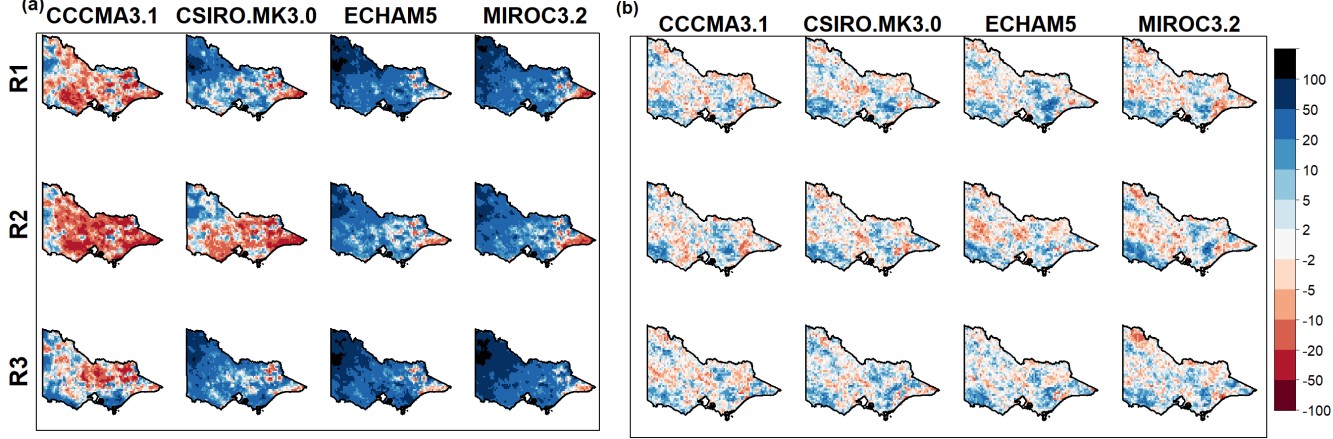

**Figure 4: Percentage bias in WRF-downscaled 99th percentile of rainfall (P99) from the GCM/RCMs indicated: (a) raw data; (b) residual bias (after bias correction).**

monthly rainfall; (c) rainfall-sequencing metrics both before and after bias correction. The range of results represents the spread of downscaled GCM hindcast spatial averages over Victoria from the NARCliM 12-model ensemble

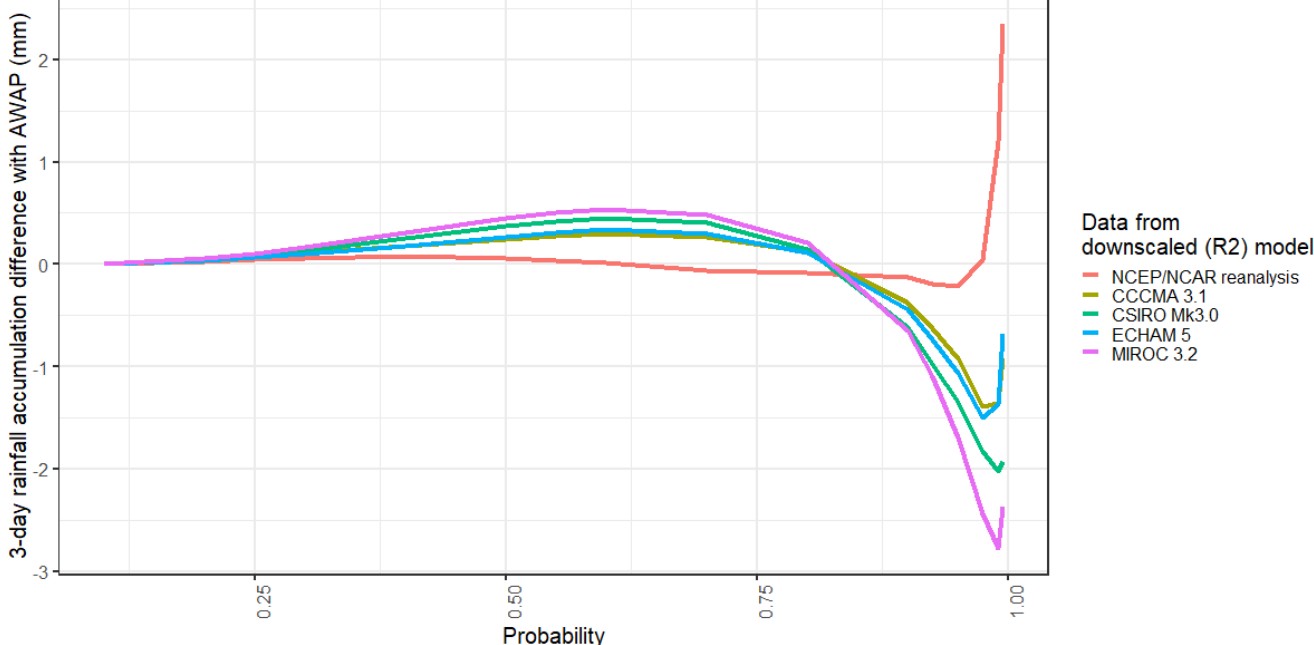

**Figure 6: Biases in spatial average 3-day rainfall accumulation percentiles. Here a 3-day moving average filter was applied to each hindcast timeseries and equivalent quantiles taken at increasing probability values (x-axis).**

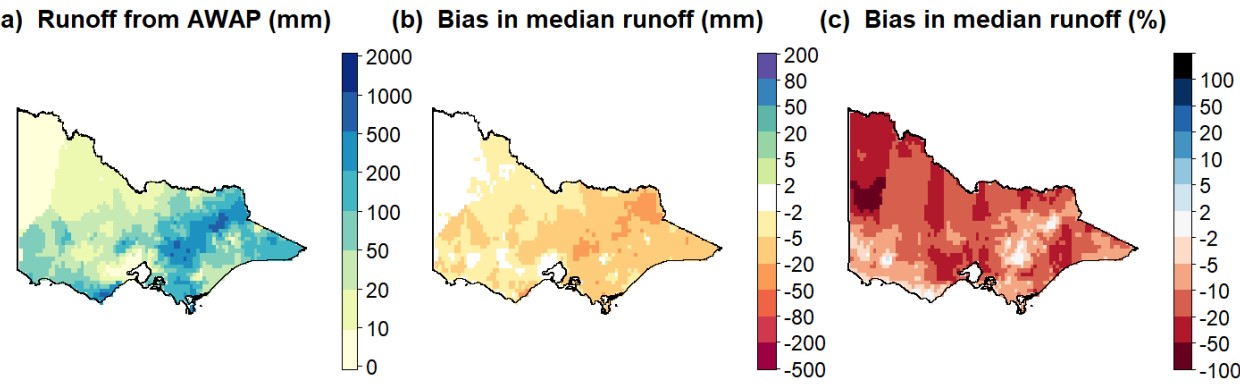

**Figure 7: Ensemble median modelled runoff over Victoria: (a) from AWAP historical observations; (b) absolute bias (mm); (c) percentage bias (%).**

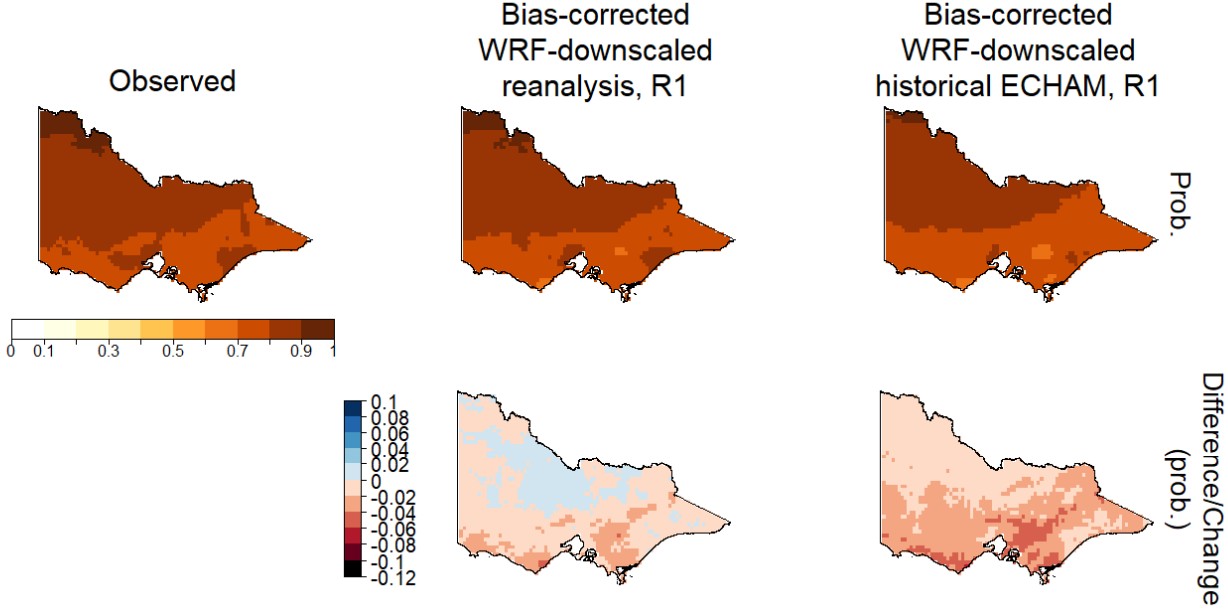

**Figure 8: Dry-dry transition probabilities (1mm threshold)**

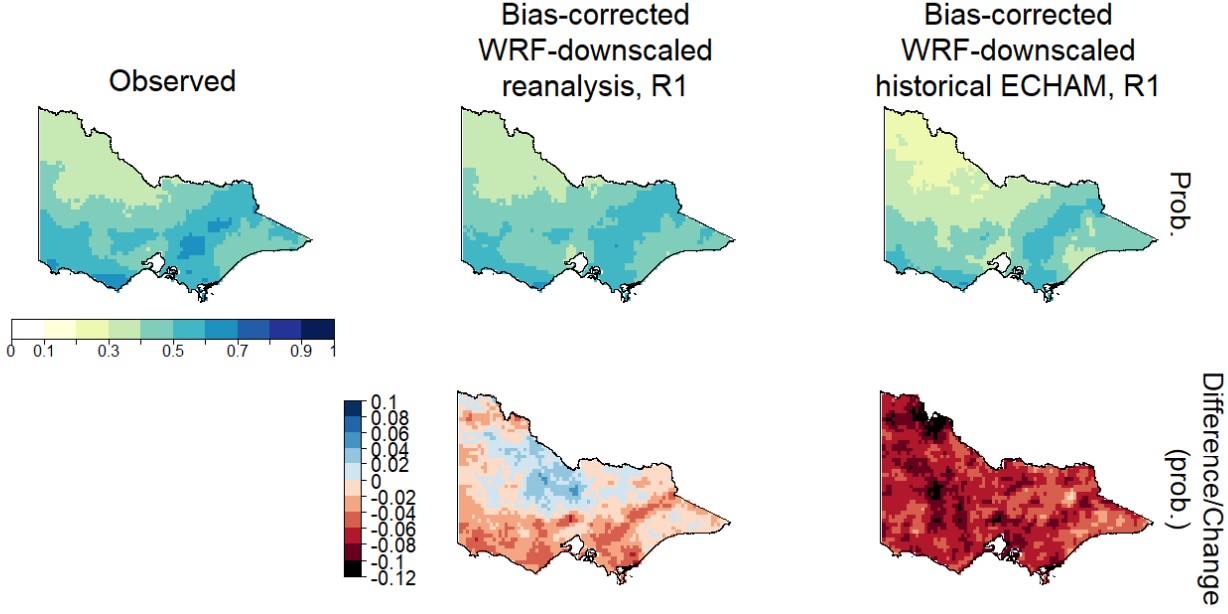

**Figure 9: Wet-wet transition probabilities (1mm threshold)**

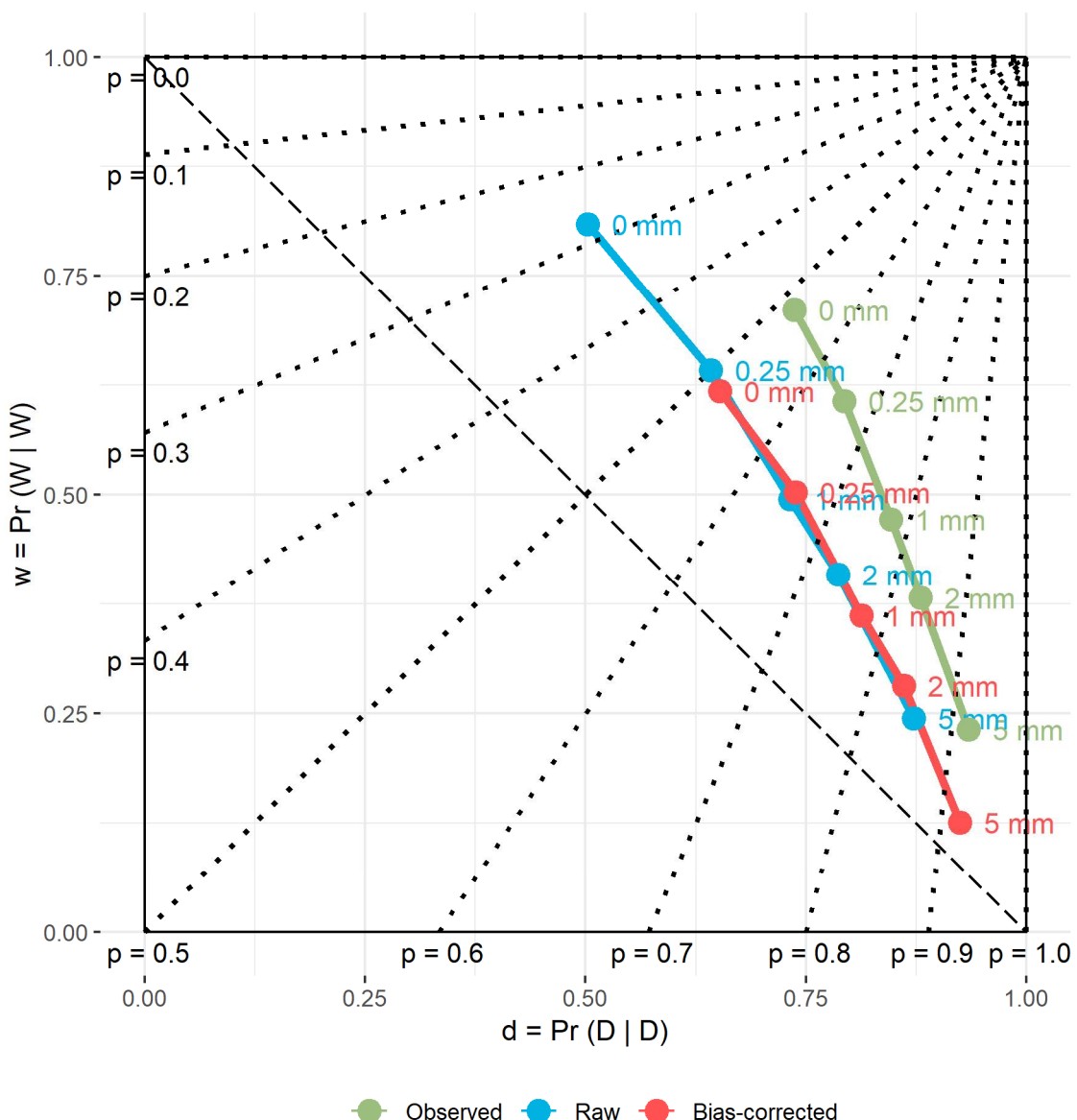

**Figure 10: An alternative perspective on quantile-quantile mapping: daily rainfall amounts and associated probabilities plotted in *d-w* space. Quantile-quantile mapping bias correction (red) maps daily rainfall amounts from the raw data (blue) to the probability contours (dotted lines) corresponding to the appropriate observed daily amount (green). The dashed diagonal line represents *p=d* and hence an independent series of events (see sect. 2.4). Points lying above or to the right represent quantiles with greater (more persistent) autocorrelation.**

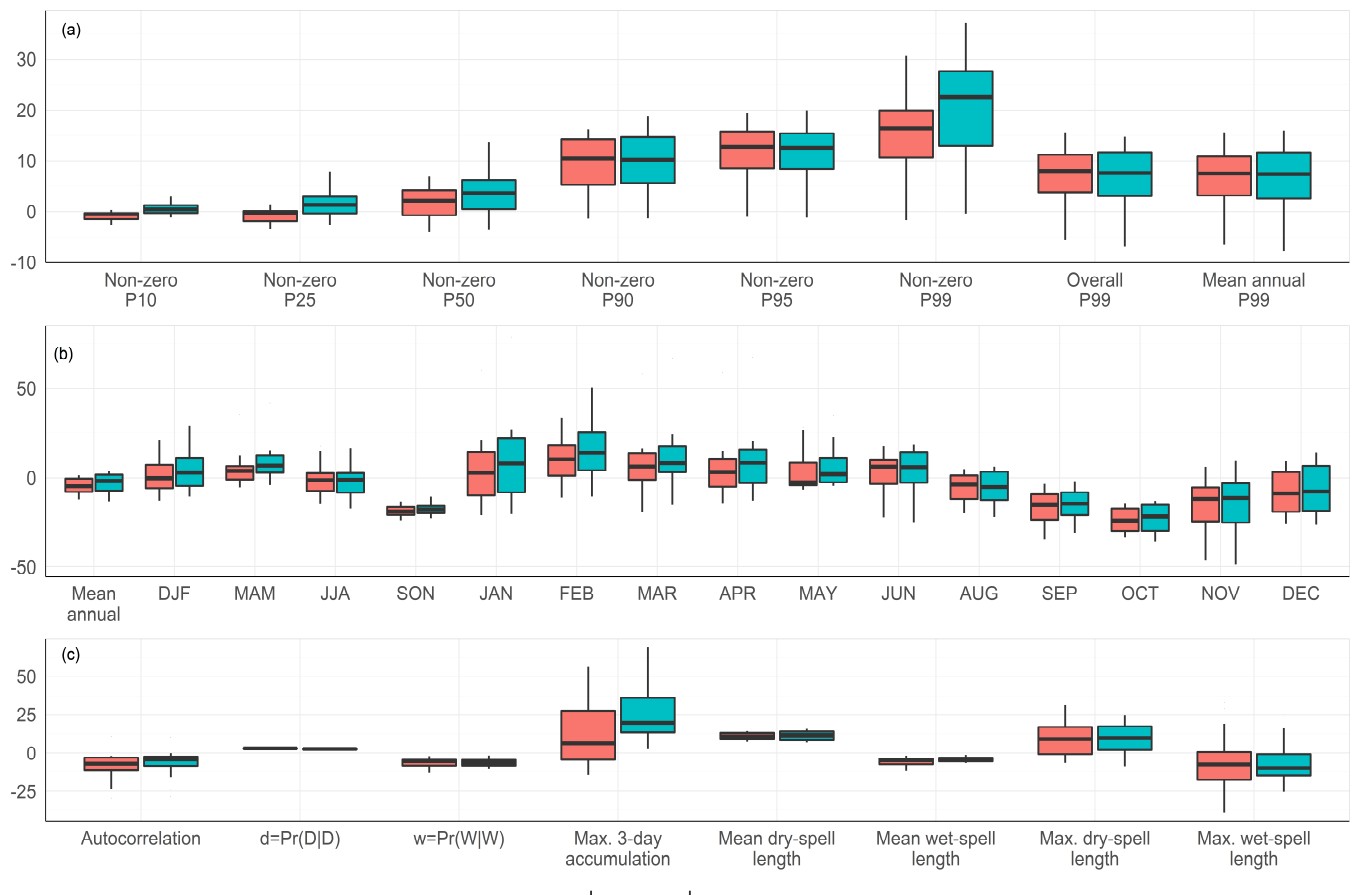

**Figure 11: Change signal (percentage difference of RCM future relative to RCM historical) in (a) rainfall percentiles; (b) mean annual, seasonal and monthly rainfall; (c) rainfall-sequencing metrics both before and after bias correction.**

