# Peer review of "Bias in downscaled rainfall characteristics"

_Hydrology and Earth System Sciences, 2019_

## Referee Comment (RC1) · Anonymous Referee #1 · 17 May 2019

The paper is well-written and concisely describes methods, results and conclusions. It analyses the effects of applying quantile-quantile mapping to precipitation data in Victoria, Australia, and exemplarily shows the method's possibilities and raises aware­ness of deficiencies. As addition to quantile mapping, lag-one transition probabilities are incorporated in order to account for rainfall sequencing. In the study, the results of quantile-quantile mapping are analysed for properly correcting the distribution of the data. It is pointed out that rainfall occurrences are represented only insufficiently not being able to properly represent patterns of consecutive rainy or dry periods that are crucial for the local water availability. Furthermore, the variability range of the raw and bias corrected RCM precipitation data and the relative magnitude of climate change signals are analysed. The article presents an interesting and valuable case study that needs revisions regarding the embedding of the study's findings in the scientific state of the art. Furthermore, the selection of figures should be reconsidered and finally

presented in a more elaborated condition.

MAJOR ISSUES

The discussion should be revised thoroughly comparing the study's results to other related studies outside the group and embedding them within the scientific state of the art. Maybe Cannon et al. (2015), Chen et al. (2013) or Maraun (2013) could be useful, as they also assessed the outcome of quantile mapping of precipitation data in Europe and Northern America or, for Australia, Agrüeso et al. (2013), Lockart et al. (2014) and Bennett et al. (2014), that you have mentioned in your introduction. Teutschbein & Seibert (2012) or Maraun (2016) could be useful for 'bias correcting climate change simulations' just to quickly name some suggestions.

MINOR ISSUES

p.1, l.21: '. . . any quantile mapping bias correction method is . . .'

p.1, l.26: '. . . Of most interest are possible changes. . .'

p.1, l.28: '. . . the spatial resolution of these models is too coarse. . .'

p.2, l.9: '. . .Teutschbein and Seibert. . .'

p.5, l.14: (Addor and Seibert, 2014) in brackets

p.5, l.18ff: Make sure to introduce all variables properly (Pr, P, p, upper/lower case D/d/W/w).

p.5, l. 26/l.29: Figure 1 and Figure 2. Later on you are often using the abbreviation Fig. only. Maybe you want to decide for one format consistently. Same inconsistency for Section/Sect.

p.7, l.7: 'Sect. 3.3.1'. Maybe 1 should be deleted here.

p.8, l.22: Close the bracket.

Maybe Sect. 3.1 could be moved closer to Sect. 3.4 or they could even be combined?

p.10, l.5: Charles et al., submitted meanwhile.

p.10, l.29: 'seasonal changes are more like half of the bias in seasonal averages'. To me, using 'like' here, sounds a bit unclear and colloquial, please rephrase.

Some of the graphs and their placements are still in a quite raw condition. I assume this is going to be revised in the final version (e.g. labels and units of colour bars, legends, cut off axis labels, maybe adapting the range of x-axis (Fig. 14-16, 6-8?), full stops at the end of each caption). Generally, I think 16 figures is a lot. Do you really need all of them to convey the message of the paper? Maybe Fig. 2 could be skipped, as it is part of Fig. 13? Or Fig. 5 could be moved to the appendix as it is supportive to another point? Or Fig. 14-16 and Fig. 6-8 respectively could be combined in one figure?

Fig. 9: Is it necessary to add the smaller steps at the higher percentiles on the x-axis? I found it less intuitive at first glance.

Fig. 10/11: You could consider adjusting the colour bars, as no blue colour appears.

---

## Referee Comment (RC2) · Anonymous Referee #2 · 20 May 2019

The main objective of the article is the analysis of how bias correction (BC) affects rainfall characteristics in a 12-member WRF-ensemble (NARCLiM simulations) forced by reanalysis and pre-selected GCMs. In particular, the authors investigate whether BC modifies the temporal persistence of dry/wet spells, transition probabilities and simulated climate change signals. Precipitation fields with realistic rainfall amount and persistence are crucial for runoff modelling and BC is thus a commonly applied technique. The paper nicely shows that the chosen BC method (empirical quantile mapping) corrects for the biases in the mean and variability, but it does not alter the temporal structure of the precipitation field. The introduced diagnostic for transition probabilities is interesting and valuable in judging the quality of simulated precipitation fields. The authors further show that the climate change signals of precipitation characteristics are mostly unaffected (with some exceptions) by the BC. Although the paper is well written, it has some major deficiencies that have to be revised in order to improve the paper.

In summary, the paper has to point to a knowledge gap the authors want to address and frame the current state and consensus of BC research, which has advanced compared to what the authors describe, in that it became much more critical (Maraun et. al, 2017). Thus, the construction and evaluation of the BC has to be better described, already beginning with the credibility of the climate models. Finally, the models (WRF and the rainfall-runoff model GR4J) need to be described in more detail. Overall, I encourage the authors to take up the points I will elaborate below.

**Major comments**

- The study must be better embedded in the current state of bias correction research. Therefore, the Introduction needs to be revised in order to better frame the current edges of bias correction, which has advanced compared to what is written by the authors. Another issue in this context is the somewhat unclear research question of the authors. The research question should be stated much more clearly, immediately after the opening, to define the overall thrust of the study. At present, the research question is weak (post-processing WRF output by a well known method and see what happens) and only appears at the end of the Introduction, after the reader went through long descriptions. Sharpening the research question is important because it is unclear where the authors see a knowledge gap they would like to close/target with their study. Recent discussion on bias correction research became much more critical on using BC as a one-tool-does-it-all or Swiss-Army Knife (Maraun et al., 2017; Maraun 2016), also the very basics of applying BC are questioned (Ehret et al., 2012). Therefore, I think the conclusion on P11L16 "..., a different approach to bias correction is needed ..." maybe points into the wrong direction, because a BC should not modify the transition probabilities, trends or climate change signals when the simulated regional climate (change) is credible. It does not mean that BC should not be used

at all, but that the whole procedure has to be viewed more critical and applied with great care. That said, the authors should take care on what their expectation and hypothesis are; it already begins with the credibility of the climate models. When it is known that BC affects climate change signal / persistence, what is the expectation when applying BC to the WRF ensemble? So instead of stating "the aim of paper is to investigate" (i.e. collecting information), it should be "our question was" (i.e. knowledge hope to gain). For instance, one interesting question to follow might be: "How to exploit/subsample an RCM-ensemble to produce the most credible bias-corrected precipitation fields for runoff modelling?".

• It is known that BC modifies the climate change signals or trends (e.g. Hagemann et al. (2011); Gutjahr and Heinemann (2013); Dosio et al. (2016)), which is why trend preserving BC methods have been developed (e.g. Switanek et al., 2017), so that the user can choose how to modify trends or change signals. In particular, if the simulated regional climate change is credible, then Maraun and Widmann, (2018, p.199) recommend to use a trend preserving BC instead of the standard quantile mapping.

• Persistence and BC is a more difficult field, but most BC methods do not affect the temporal structure of the data (persistence), so that the corrected fields retain the inherit persistence of the driving model (Maraun et al., 2017; Maraun and Widmann, 2018). In contrast, however, Rajczak et al. (2016) demonstrate the opposite, although artefacts might be introduced (Maraun and Widmann, 2018, p. 182). This discussion should be taken up by the authors.

• I am not sure if I correctly understood where the 12-member WRF ensemble originates from. I suppose Evans et al. (2014) chose this subsample from the 36-member WRF ensemble, described by Evans et al. (2013), based on their skill and independence. Thereby the skill of the WRF simulations was based on how they simulate two-week periods centred on eight storm events. My concern is in line with the remarks by Evans et al. (2014) in that it is unknown how skillful the simulations are at climate time scales, when the long-term memory becomes important, e.g. for dry spells. As stated by Maraun et al. (2017), bias correction is only partly a statistical issue, but already begins with the selection of credible climate models. If these pre-selected 12 WRF simulations already have fundamental errors in precipitation temporal structure (e.g. because of misplaced circulation or misrepresented weather-types), then it cannot be expected that BC corrects the persistence (and probably should not). Maybe a different set of GCMs and subsequent WRF simulations are better suited as input for runoff modelling. I recommend discussing how suited the 12 WRF simulations are for runoff modelling in Victoria on the climate scale. In this sense, maybe the number of WRF simulations considered for bias correction and eventually for runoff modelling could be reduced to only the most credible ones. Although I understand that the authors want to show the range of uncertainty from the WRF ensemble, in practice one does not want to use RCM simulations that have fundamental errors (e.g. in circulation, precipitation persistence, etc.) for runoff modelling, so these could be discarded from the analysis in advance.

- It should be discussed how bias-corrected precipitation fields might be inconsistent with other fields of the WRF model (e.g. temperature). BC reduces the precipitation volume error but might create other problems for impact models due to inconsistencies with other atmospheric fields. The problem here is that BC is not intended to modify the persistence and it mostly inherits the temporal structure of the driving model. It does, however, implicitly modify the transition probabilities as it adjusts the number of wet days (the wet day threshold is a percentile of the distribution). Maraun et al. (2017) and Maraun and Widmann (2018) argue that if the persistence of the driving GCM or RCM is all too wrong, then this model should not be used for downscaling and it is not recommended to apply a bias correction (which the authors also state on P2 L27). Errors in persistence originate mainly

from wrong dynamics in the climate models, so that specific circulation features are misplaced or misrepresented. These errors in dynamics then manifest themselves in errors of transition probabilities or as seasonal biases. For instance, the WRF models used here simulate too much precipitation throughout the year, but mostly so in January – May. Could this indicate fundamental errors in the circulation of the models in these months? As outlined above, a BC cannot overcome this problem and is thus not recommended (Maraun et al., 2017). If a BC is still applied, it causes inconsistencies with the other fields. Suppose for instance, if the precipitation fields were corrected to match longer dry spells, then the large-scale synoptics would still simulate wet conditions, which causes inconsistencies and might produce errors/artefacts in runoff models. The authors should discuss these issues and whether these inconsistencies affect the runoff model.

- The evaluation strategy of the bias correction method is unclear. BC evaluation is a difficult task (Maraun et al., 2017) and has to be done very carefully. The important part here is how the BC performs for data outside the calibration period. As QM may be prone to overfitting, in particular for the tails, such an evaluation is important to understand for instance why the 'Non-zero P99' distribution in Fig. 14 is corrected to higher values, which might also be relevant for why the DJF and MAM change signals of annual mean precipitation increase after BC (Fig. 15). Maybe a Q-Q plot of a representative time series of the raw and corrected precipitation might help to understand how the BC performs (and whether it is too flexible/stiff) for data not used to construct the transfer functions. This might also help to understand why the rainfall is corrected to too low values in Feb, Apr, May, and June (P6, L31f) (issue of overfitting?). Was it tested whether a seasonal correction is the best strategy? There are papers that analyse time scales issues for bias correction (e.g. Haerter at al., 2011; Reiter et al., 2018), concluding that the biases are not independent of the timescale and might introduce errors if assumed so. Has something similar been tested, or what is the argument for a

seasonal correction?

- Section 2 lacks a detailed model and data description. Mention that a WRF-ensemble consisting of 12 members was used (except you decide to reduce number of members). By which data set was the WRF hindcast simulation forced? I suppose from NCEP (Phase 1 described by Evans et al. (2014)), but why does ERA-I later appear in the legends of Fig. 11 and 12?. Define here also what R1, R2, R3 means, and what the main differences between these three WRF configurations are. It should be mentioned whether these 12 WRF simulations (plus the hindcast) were evaluated on the climate scale for the Victoria state, as mentioned before. If not, it should be at least included in the Discussion. Furthermore, there is no description of the runoff model GR4J, it is suddenly mentioned in section 3.2. A description should be included in section 2 (also mention which atmospheric input fields are used to force GR4J, inconsistency issue).

**Minor comments**

- The authors may consider to revise the title of the paper to account for the runoff aspect.

- Also, consider to reduce the number of figures and correct the figure legends (be clear that WRF simulations are referred to, not the GCMs). Also add more explanations to the figure captions (abbreviations, before/after BC, add data set used to construct the figure, e.g. Fig.7 and similar). I further suggest to enlarge the axis annotations.

- P3, L12 and L18: Better refer to parametric and non-parametric distributions, because also empirical distributions are 'distributions'.

- P3, L16ff: Does this statement refer to corrections on the calibration data, or when applied to data outside the calibration period? Overly flexible methods might introduce artifacts at the tails outside the calibration period. Inappropriate (too stiff) parametric methods may introduce unrealistically high values, when much higher values than observed appear in a future scenario (Volosciuk et al., 2017) because of the extrapolation.

- P3, L23: How was this evaluated? Again, only on the calibration data or also on data outside the calibration period? For the calibration data, an almost perfect correction results by design by all methods.

- P4, L16ff: Evans et al. (2014) is not the correct reference for how WRF reads in lateral/lower boundary data from a GCM. Put this reference to the next sentence: "The NARCLiM projections (Evans et al., 2014)..."

- P5, L26 and L29: correct 'Figure' to 'Figure 2.' (you might want to drop this figure, as the information is already in Fig. 13).

- P6, L.16f: Only if it is assumed that the bias is time-independent, otherwise it was demonstrated that the chosen timescale for constructing the transfer functions impacts for instance the annual mean (Haerter et al, 2011; Reiter et al., 2018).

- P6, L27: remove "after bias correction".

- P6, L31: effect of overfitting? Would this also occur if a monthly BC is used instead of a seasonal?

- P7, L6: maybe better use "considerably" (or similar), "significantly" has a connotation related to statistical testing.

- P7, L8: Be more precise: Instead of "This results .." write e.g. "Due to good performance in correcting dry-dry transition probabilities, the bias in mean and maximum dry spells is well corrected, ....".

HESSD

Interactive
comment

- P7, L10 and Fig.9: Does Figure 9 show the 3-day rainfall bias of the WRF simulations (text) or of the GCMs (legend)? And then, before or after the bias correction?

- P7, L18: Explain what PET means; if it is a data set, then describe it in section 2.

- P7, L27: Give more than one reference for "a number of studies..".

- P8, L3: Be precise which models are used where. Sometimes reanalysis and GCM results are wrongly used, when WRF models forced by either NCEP or a GCM are meant. There are other places in the manuscript as well (e.g. legend of Fig. 9, or titles of Fig. 11 and 12).

- P8, L8: Add "... wet-wet transition probabilities are largest over the high-runoff producing region (Fig. 12),...".

- P8, L32: replace "northeast" with "to the right".

- P9, L18: Replace "Fig.16" by "Figure 16".

- P10, L4f: Replace "Charles et al. (2019, submitted). (Or insert reference if already published).

**References:**
Dosio et al. (2016), doi: 10.1002/2015JD024411.
Ehret et al. (2012), doi: 10.5194/hess-16-3391-2012.
Gutjahr and Heinemann (2013), doi: 10.1007/s00704-013-0834-z.
Haerter et al. (2011), doi: 10.5194/hess-15-1065-2011.
Hagemann et al. (2011), doi: 10.1175/2011JHM1336.1.
Maraun et al. (2017), doi: 10.1038/NCLIMATE3418.
Maraun and Widmann (2018): Statistical Downscaling and Bias Correction for Climate

Research. Cambridge University Press, ISBN: 978-1-107-06605-2.

Rajczak et al. (2016), doi: 10.1175/JCLI-D-15-0162.1.

Reiter et al. (2018), doi: 10.1002/joc.5283.

Switanek et al. (2017), doi: 10.5195/hess-21-2649-2017.

Volosciuk et al. (2017), doi: 10.5194/hess-21-1693-2017.
* * *

---

## Referee Comment (RC3) · Anonymous Referee #3 · 5 Jun 2019

The paper is well written, and I agree that bias correction methods can and should be improved by correcting model bias with regards to temporal persistence. The authors have done a good job in their attempt to effectively remove modeled temporal bias at individual grid cells. That said, the authors need to provide better context with respect to the most recent advancements in bias correction methods. This was pointed out by the other reviewers as well, and they have provided some of the relevant literature that should be cited and discussed in your paper.

Some major points:

1. Yes, as the other reviewers have already stated, please place the application of bias correction in a broader context. Should we even bias correct rainfall or temperature data to force a hydrological model (Ehret et al., 2012, see Referee 2)? Ehret et al., (2012) argue that perhaps bias correction is simply applied to the end of the modeling

chain, and as a result, streamflow values themselves are bias corrected. Personally, I do not agree with this argumentation, because establishing an error correction function that estimates the bias of the hydrological model using observed forcings will not translate well to climate model forcings (which are themselves biased). The highly non-linear response of streamflow to biased modeled precipitation could prove problematic. In any event, you need to discuss some of these issues related to applying bias correction methods to climate model output.

Similarly, please add more discussion concerning the "inflation issue" or "non-stationarity" of quantile mapping. This is not a trivial component to methods such as the standard quantile mapping that assume a stationary error correction function. This is especially important when attempting to draw conclusions about future model projected changes to meteorological variables. What methods have been proposed to handle the inflation issue? (Cannon et al., 2015; Switanek et al., 2017; see Referee 2).

2. There are many approaches that can be taken to improve upon existing bias correction methods. The authors have tried to tackle an important one: that of rainfall persistence in time. The paper needs to discuss some of the other shortcomings of a method like quantile mapping, and why they chose to focus solely on improving temporal persistence. There are two other obvious deficiencies of traditional quantile mapping that will impact your results. The first is related to the inflation issue and the assumption of a stationary error correction function. This was highlighted above. However, even more closely related to your issue of temporal persistence, is that of spatial persistence. Your goal is to have precipitation events that are more realistic in their persistence in time (when compared to observations) to force a hydrological model. Too many wet days or too many dry days, statistically speaking, will be exacerbated when routed through non-linear streamflow response to precipitation, and this can ultimately lead to incorrect conclusions about how the hydrology is changing. Equally important with respect to streamflow output is spatial persistence (Bardossy and Pegram, 2012). Consider an example where average observed events cover 20 connecting grid cells, for a particular season, and a particular model, on average, cover 40 connecting grid cells. In this example, the model is putting down rain across a greater extent, and this will inflate the tail of the hydrological extreme events. I would argue that this effect has at least as strong of an impact on hydrologic response as temporal persistence. Bardossy and Pegram (2012) present a method to recorrelate model data so that the extent of events, on average, are comparable to observed event sizes. The authors need to either include some additional analysis concerning spatial persistence, or at the least, they need to discuss the contribution that this could have on streamflow output. I realize that you have compared the "spatial rainfall fields" from the models to that of "observed mean annual rainfall" (page 6, line 5). This is different than what I am pointing out. Averaging across days and seasons can hide differences between modeled and observed cross-correlations.

Some minor points:

Figure 3: It could be helpful for the reader to place Victoria geographically. Maybe you want to have a subplot in Figure 3 outlining Victoria in Australia.

Page 4, line 18: It seems that a combination of both CMIP3 and CMIP5 data have been used. Why use CMIP3 at all? CMIP5 has been around for quite some time now. This paper is not trying to show the improvement, or lack thereof, in model performance between CMIP3 and CMIP5.

Page 5, line 26: Figure 2 instead of "Figure"

References:

Bardossy, A., and Pegram, G., Multiscale spatial recorrelation of RCM precipitation to produce unbiased climate change scenarios over large areas and small, Water Resources Research, 48, W09502, 2012.

---

## Author Comment (AC1) · 11 Jul 2019

**Authors' comment**

We thank the reviewers for their thorough and helpful comments on the manuscript. In replying to the reviewers, we have collated the response into six section as follows:

1. Sharpening of the research question
2. Better description of NARCliM models and data in section 2 of the manuscript
3. Addressing the credibility of underlying climate models
4. Assessing the effect of bias correction on change factors in the future (i.e. whether to select bias correction methods that preserve trends)
5. Inflation and maintaining spatial correlation for runoff modelling
6. Consistency between rainfall and other atmospheric variables

More generally, for the revised version of the manuscript, we plan an updated literature review (encompassing the references and discussion points below) for the introduction, an updated discussion section (again using the response below), a critical review of the figures including reduction in the total number, and an updated evaluation of the bias correction method in line with referee #2's comment. Finally, we thank the reviewers for their minor comments and will incorporate these in the revised version.

**1 Sharpening of the research question**

[revised manuscript text omitted]

a quantitative cold and wet bias generally (Ji et al., 2016). In analysing the RCM component (WRF) of NARCliM, Evans and McCabe (2010) conclude that the El Niño-Southern Oscillation, the chief climate process modulating interannual variability of rainfall (Power et al., 1999), was well modelled over south-eastern Australia. Evans and McCabe (2010) also concluded that the severity and duration of recent prolonged droughts over south-eastern Australia were also captured, although the spatial pattern was not characterised exactly. The sub-tropical ridge, which determines the seasonal positioning of storm tracks over southern Australia, was less well represented by WRF (Andrys et al., 2016). For hydrological applications, a specific combination of land and atmospheric circulation schemes (R2) is recommended for hydrological applications (Olson et al., 2016). Although we consider the entire modelling ensemble in this paper, in a related paper (Charles et al., submitted) we use only that specific RCM physics scheme for modelling runoff. Based on the results cited above, we have confidence in the modelling setup of NARCliM to represent atmospheric circulation for southern Australia reasonably well, although we acknowledge that bias correction of NARCliM for end-user applications should consider model skill in atmospheric circulation.

**4 Assessing the effect of bias correction on change factors in the future (i.e. whether to select bias correction methods that preserve trends)**

[revised manuscript text omitted]

---

## Author Response (AR1)

**Bias in downscaled rainfall characteristics**

Notes on revision for referees and editor

We once again thank the referees and editor for their very helpful reviews and suggestions. We have addressed the major issues in the previous authors' comments uploaded to the Interactive Discussion. In response to these major comments, we have

5   adjusted the manuscript where appropriate (introduction, discussion as well as relevant places throughout). Here we directly address the remaining referee comments

**Anonymous Referee #1**

p.1, l.21: '... any quantile mapping bias correction method is ...'

| done |
| --- |

10   p.1, l.26: '... Of most interest are possible changes...'

| done |
| --- |

p.1, l.28: '... the spatial resolution of these models is too coarse...'

| done |
| --- |

p.2, l.9: '...Teutschbein and Seibert...'

15   | done |
| --- |

p.5, l.14: (Addor and Seibert, 2014) in brackets

| done |
| --- |

p.5, l.18ff: Make sure to introduce all variables properly (Pr, P, p, upper/lower case D/d/W/w).

20   | This section has been revised and reworded so that it should be clear that $D_n$ and $W_n$ refer to the nth day being dry or wet respectively. |
| --- |

p.5, l. 26/l.29: Figure 1 and Figure 2. Later on you are often using the abbreviation Fig. only. Maybe you want to decide for one format consistently. Same inconsistency for Section/Sect.

| We have double checked figure and section references and they are consistent with the HESS |
| --- |
25   | style guide (i.e. abbreviated to Fig./Sect. except when starting a sentence). |

p.7, l.7: 'Sect. 3.3.1'. Maybe 1 should be deleted here.

| done |
| --- |

p.8, l.22: Close the bracket.

| done |
| --- |

30   Maybe Sect. 3.1 could be moved closer to Sect. 3.4 or they could even be combined? p.10, l.5: Charles et al., submitted meanwhile.

| We prefer section 3.1 and 3.4, but acknowledge that the section title is not representative. Section 3.1 is intended to show regional, spatial information, whereas section 3.4 is spatially averaged. We rename section 3.1 to "Assessment of Regional Performance of Modelled |
| --- |
35   | Rainfall". |
| |
| The Charles et al. paper has been submitted to HESS and the reference updated. |

p.10, l.29: 'seasonal changes are more like half of the bias in seasonal averages'. To me, using 'like' here, sounds a bit unclear and colloquial, please rephrase.

40   | "more like" replaced with "closer to" |
| --- |

Some of the graphs and their placements are still in a quite raw condition. I assume this is going to be revised in the final version (e.g. labels and units of colour bars, legends, cut off axis labels, maybe adapting the range of x-axis (Fig. 14-16, 6-8?), full stops at the end of each caption). Generally, I think 16 figures is a lot. Do you really need all of them to convey the message of the paper? Maybe Fig. 2 could be skipped, as it is part of Fig. 13? Or Fig. 5 could be moved to the appendix as it is supportive to another point? Or Fig. 14-16 and Fig. 6-8 respectively could be combined in one figure? Fig. 9: Is it necessary

45   to add the smaller steps at the higher percentiles on the x-axis? I found it less intuitive at first glance.

| The figures have been reworked in in some cases combined as suggested. |
| --- |

Fig. 10/11: You could consider adjusting the colour bars, as no blue colour appears.

**Anonymous Referee #2**

- The authors may consider to revise the title of the paper to account for the runoff aspect.

> We prefer to leave the title as is, as the companion paper (Charles et al., 2019) deals directly with runoff

- Also, consider to reduce the number of figures and correct the figure legends (be clear that WRF simulations are referred to, not the GCMs). Also add more explanations to the figure captions (abbreviations, before/after BC, add data set used to construct the figure, e.g. Fig.7 and similar). I further suggest to enlarge the axis annotations.

> As above, all figures have been redrawn to be clearer encompassing the referee #2's suggestions

- P3, L12 and L18: Better refer to parametric and non-parametric distributions, because also empirical distributions are 'distributions'.

> We have adjusted this but note that non-parametric distributions are not necessarily empirical so retain 'empirical' in parentheses.

- P3, L16ff: Does this statement refer to corrections on the calibration data, or when applied to data outside the calibration period? Overly flexible methods might introduce artifacts at the tails outside the calibration period. Inappropriate (too stiff) parametric methods may introduce unrealistically high values, when much higher values than observed appear in a future scenario (Volosciuk et al., 2017) because of the extrapolation.

> This was presented in a split-sample experiment by Teng et al. (2015) and a citation has been provided.

- P3, L23: How was this evaluated? Again, only on the calibration data or also on data outside the calibration period? For the calibration data, an almost perfect correction results by design by all methods.

> This was presented in a split-sample experiment by Teng et al. (2015) and a citation has been provided.

- P4, L16ff: Evans et al. (2014) is not the correct reference for how WRF reads in lateral/lower boundary data from a GCM. Put this reference to the next sentence: "The NARCLiM projections (Evans et al., 2014)…"

> Done

- P5, L26 and L29: correct 'Figure' to 'Figure 2.' (you might want to drop this figure, as the information is already in Fig. 13).

> Updated. Note figure 2 has been dropped and is included in figure 10 as suggestd.

- P6, L.16f: Only if it is assumed that the bias is time-independent, otherwise it was demonstrated that the chosen timescale for constructing the transfer functions impacts for instance the annual mean (Haerter et al, 2011; Reiter et al., 2018).

> Correct, and we refer here to historical bias correction. The quantiles themselves are designed to be corrected exactly but we are highlighting here that P99 is not corrected exactly due to interpolation.

- P6, L27: remove "after bias correction".

> Done

- P6, L31: effect of overfitting? Would this also occur if a monthly BC is used instead of a seasonal?

> Added "this could be reduced to zero using monthly correction factors, although at the potential cost of overfitting".

- P7, L6: maybe better use "considerably" (or similar), "significantly" has a connotation related to statistical testing.

> Significantly replaced with considerably

- P7, L8: Be more precise: Instead of "This results .." write e.g. "Due to good performance in correcting dry-dry transition probabilities, the bias in mean and maximum dry spells is well corrected, …".

> Replaced "this results" with "Since dry spells are directly related to dry-dry transition probabilities,"

- P7, L10 and Fig.9: Does Figure 9 show the 3-day rainfall bias of the WRF simulations (text) or of the GCMs (legend)? And then, before or after the bias correction?

> Figure 9 is calculated on bias-corrected downscaled GCM time series, legend and caption updated accordingly.

- P7, L18: Explain what PET means; if it is a data set, then describe it in section 2.

> PET is spelt out (potential evapotranspiration). This part of the paper is taken from Charles et al. (2019), so rather than include details here we have provided an extra citation.

- P7, L27: Give more than one reference for "a number of studies..".

| Additional references have been provided here |
| --- |

- P8, L3: Be precise which models are used where. Sometimes reanalysis and GCM results are wrongly used, when WRF models forced by either NCEP or a GCM are meant. There are other places in the manuscript as well (e.g. legend of Fig. 9, or titles of Fig. 11 and 12).

| We have updated these throughout the manuscript, as well as changing WRF in some instances to NARCliM when appropriate. |
| --- |

- P8, L8: Add "... wet-wet transition probabilities are largest over the high-runoff producing region (Fig. 12),...".

| included |
| --- |

- P8, L32: replace "northeast" with "to the right".

| Done |
| --- |

- P9, L18: Replace "Fig.16" by "Figure 16".

| "Fig. 16" remains as it is consistent with HESS style |
| --- |

- P10, L4f: Replace "Charles et al. (2019, submitted). (Or insert reference if already published).

| Reference updated |
| --- |

**Anonymous Referee #3**

- Figure 3: It could be helpful for the reader to place Victoria geographically. Maybe you want to have a subplot in Figure 3 outlining Victoria in Australia.

| Figure 3 (now Fig. 2) has been updated with an inset subplot showing Victoria in relation to Australia. |
| --- |

- Page 4, line 18: It seems that a combination of both CMIP3 and CMIP5 data have been used. Why use CMIP3 at all? CMIP5 has been around for quite some time now. This paper is not trying to show the improvement, or lack thereof, in model performance between CMIP3 and CMIP5.

| The available NARCliM data is downscaled from CMIP3 data only; CMIP5 models are currently being downscaled by NARCliM modellers. As mentioned in our previous response to reviewers (and included in the discussion section of the revised manuscript), using dynamically downscaled data for hydrological projections is largely opportunistic, relying on currently available data. Ideally, we would use CMIP5 downscaled data but this is not currently available. We intend for the findings of this paper to be applicable to bias correction for CMIP5 and CMIP6 dynamically downscaled model data. We have included a citation to the future NARCliM work in the relevant place. |
| --- |

- Page 5, line 26: Figure 2 instead of "Figure"

| Done |
| --- |

The revised manuscript follows:

[revised manuscript text omitted]
̶: (a) rainfall percentiles ̶b̶e̶f̶o̶r̶e̶ ̶a̶n̶d̶ ̶a̶f̶t̶e̶r̶; (b) mean annual, seasonal and monthly rainfall; (c) rainfall-sequencing metrics both before and after bias correction. The range of results represents the spread of GCM hindcast spatial averages over Victoria from the NARCliM ̶W̶R̶F̶ 12-model ensemble**

[Figure]

Figure 7: Relative bias in annual, seasonal and monthly averages

[Figure]

Figure 8: Relative bias in rainfall sequencing related metrics

[Figure]

[Figure]

5   **Figure 96: Biases in spatial average 3-day rainfall accumulation percentiles. Here a 3-day moving average filter was applied to each hindcast timeseries and equivalent quantiles taken at increasing probability values (x-axis).**

[Figure]

**(a) Runoff from AWAP (mm)**    **(b) Bias in median runoff (mm)**    **(c) Bias in median runoff (%)**

**Figure 7: Ensemble median modelled runoff over Victoria: (a) from AWAP historical observations; (b) absolute bias (mm); (c) percentage bias (%).**

[Figure]

**Figure 811: Dry-dry transition probabilities (1mm threshold)**

[Figure]

**Figure 912: Wet-wet transition probabilities (1mm threshold)**

[Figure]

[Figure]

**Figure 10<s>13</s>: An alternative perspective on quantile-quantile mapping: daily rainfall amounts and associated probabilities plotted in *d-w* space <s>(cf. Fig. 2)</s>. Quantile-quantile mapping bias correction (red) works by map<s>pings</s> daily rainfall amounts from the raw data (blue <s>curve</s>) to <s>to</s> the probability contours (<s>dashed</s>dotted lines) corresponding to the appropriate observed daily amount (green).**

[Figure]

**Figure 11̶4̶: Change signal (percentage difference of RCM future relative to RCM historical) in (a) rainfall percentiles ̶b̶e̶f̶o̶r̶e̶ ̶a̶n̶d̶ ̶a̶f̶t̶e̶r̶ ̶b̶i̶a̶s̶ ̶c̶o̶r̶r̶e̶c̶t̶i̶o̶n̶.̶; ̶(b) mean annual, seasonal and monthly rainfall; (c) rainfall-sequencing metrics both before and after bias correction.**

[Figure]

[Figure]

**Figure 16: Change signal in rainfall sequencing metrics before and after bias correction**

---

## Referee Report (RR1)

The revised version of the paper now includes some of the proposed changes from all reviewers. It has definitely improved, also in terms of reducing the number of figures, so that a clearer structure emerged. Nevertheless, in my opinion there are still some open issues – in particular concerning the expectations on bias correction (BC) / hypothesis of the analysis, model/data description, and method description. These issues are either still unclear or insufficiently presented, but are directly relevant to the research question of the paper: what is the effect of BC on rainfall characteristics relevant for runoff modelling? Therefore, I still suggest that major revisions are required. In the following, I will outline these issues.

**Major comments**

- The Introduction still lacks some relevant aspects. Most of all, the authors should state what a BC is meant to correct (amount of precipitation), what cannot be corrected (e.g. temporal errors, such as persistence of transition probabilities), and what is known to be affected by BC (change signals, persistence in some studies) and whether this effect is intended. Based on this literature review (which is still scattered over the whole text), hypotheses/expectations should be formulated that guide the analysis. For instance, it could be expected that a BC would remove the systematic errors in rainfall amounts, but would not correct the transition probabilities or 3-day precipitation totals. Since other studies found that BC affects persistence and transition probabilities, this should be a motivation for the study and for introducing the new diagnostic presented by the authors.

- As another motivation, previous results with the here used WRF R1-R3 simulations should be summarized already in the Introduction. Obvious would be the results of Jin et al. (2016), who evaluated rainfall (and temperature) of the same WRF simulations on climate time scale. Jin et al. even concluded that a bias correction is required, which renders a natural motivation for this study. I further noticed that the WRF simulations presented here actually stem from a double nesting (50km CORDEX Australasia domain and the inner 10km NARCliM domain, Fig.1 in Jin et al.). It would be good to write this in the model description – maybe also mention that it is the same simulations as in Jin et al.

- Other important evaluation results of the here used WRF simulations, such as presented on p. 9, L. 6ff of the Discussion, should be presented already in the Introduction or model description. Thereby, note that Gilmore et al (2016) only evaluated 15-day simulations focusing on a specific extreme precipitation event; thus, it is not guaranteed that WRF behaves similar at the daily scale for other synoptic situations. These evaluations should then constitute the basis from where you start your analysis. Of particular importance is the information that WRF R2 renders the best configuration for hydrological applications (p.9, L.13-15) (why?) with respect to Olson et al. (2016). This information has to go into the Introduction, or into the model/data description. A natural question arises then whether your analysis support their conclusion? Finally, make clear how the models have been selected (e.g. as it is done in the Introduction of Jin et al., 2016) and present this information in the model/data description. Only buried in the text (p.6, L.31), you state that WRF selection was based on credible simulations of 2-week heavy-precipitation periods. Do you then expect that these configurations faithfully represent other rainfall metrics, e.g. transition probabilities? If for instance transition probability/persistence of rainfall was not a selection criteria, but is important for hydrological applications, then the a-priori expectation should be that BC would not help and a possible conclusion from your results could be e.g. that hydrologically relevant metrics should be added to a GCM/RCM selection process (based on CMIP5 etc.).

- In the discussion (p.8,L.28ff) you state that WRF has relatively large rainfall biases. You then conclude that there are model errors whose origin needs to be analysed and whether they render the physics implausible. In contrast, on p.8, L.37, you write these WRF simulations have

been extensively tested, and on p.9, L.6f that there is reasonable confidence in these simulations, although they have a general cold and wet bias. So, I guess the origin of these errors is understood to some degree and was already attributed in other studies?

- On p. 8, L.2f you raise the need for bias correction methods that correct the occurrences of rainfall events – in other words, the temporal structure. Once again, temporal errors indicate fundamental model errors that cannot be corrected by BC methods (e.g. Maraun et al., 2016; Maraun and Widmann, 2018). They are directly linked to errors in the dynamics of the climate model, which could result from many different sources, e.g. from coarse topography that affects the dynamics in a wrong way or missing relevant regional processes. BC is not meant to correct these errors. In contrast, on p.2, L.24f you state: "The underlying assumption of bias correction is that the RCM output faithfully represents climate processes relevant for rainfall, although the amounts themselves may not be accurate"; and on p.3, L. 15ff: "QQM bias correction cannot remove biases in rainfall sequencing …". This implicitly implies that the temporal aspect (persistence, transition probabilities, etc.) is assumed to be realistically simulated by the RCM and should thus be part of the model evaluation procedure prior to any BC. Any attempt to correct temporal errors of the model, however, could introduce unwanted major artefacts.

- A motivation is missing why the authors repeat the bias correction with an empirical QQM, when there already is a corrected rainfall set using a double-gamma QQM (p.4, L.14f) that is provided by Evans and Argüeso (2014). Is there any reason not to use this data set? At least, provide a statement why this already existing data set was not used.

- The description of the models is still somewhat confusing. Please clearly refer to WRF simulations instead of either "bias-corrected reanalysis" or "bias-corrected GCM rainfall" (e.g. p.7, L.10f). Otherwise, it is very confusing (also in the figure labels), and it is just wrong because reanalysis/GCMs were downscaled with WRF and not directly bias corrected.

- I further recommend to revise section 2. First, all models that are used in the paper should be named here, including the runoff model GR4J that is first introduced in the results section (section 3.2 on p. 6, L.35). Then provide a short description of the WRF model with exact model version, model resolution, number of vertical levels, etc., and in particular briefly mention the differences of the WRF R1 – R3 configurations. It is still unclear to the reader whether R1 - R3 are different realisations based on slightly different initial fields or on different physics. Part of that information (e.g. model resolution, time period, etc.) is provided in section "2.2 Daily data", but that is not the correct place (maybe rename 2.2 to "Observations"). Furthermore, on p.4, L.6f, be more precise in describing that you use these three (R1-R3) WRF configurations for each of the four chosen GCMs, resulting in 12 simulations in total. One could misread p.4, L.6f that you downscale only three GCMs and that these downscaled WRF simulations are labelled R1-R3. Mention also that these result from double-nesting into the GCM/NCEP: 50km CORDEX-Australasia domain and 10km NARCliM domain. In section 2.2 only NCEP is stated as the reanalysis data used for this study, but in Fig. 8 & 9 you show results from WRF nested into ERA-Interim?

- The description of the BC method still lacks some critical information. Although a reference to Teng et al. (2015) was added to the Introduction (but not to section 2.2?), it is still not clear how exactly the QQM was constructed for the here presented WRF simulations. Teng et al. (2015) use different time periods, different observations to construct the QQM, and different WRF simulations. Apart from the different distribution, is the QQM method identical to the QQM of Evans and Argüeso (2014), as suggested on p.4, L.12 ("… which is similar in many respects to the non-parametric procedure we apply …")? Did you use the full historical period (1990-2009) to construct QQM and then applied it to the complete period? Or did you split it into calibration (dependent) and validation (independent data) subsets and then apply the so

constructed QQM to the complete period? If you did the first without validating QQM with independent data, you cannot judge how strong overfitting is. If the complete historical period was used to construct the transfer functions, then a BC always reduces the mean error to zero (neglecting linear interpolation from 'qmap'), but the overfitting only takes effect on independent data (future time period data). Thus, overfitting could be large depending on how the QQM was constructed. Without such information, it is not possible to repeat the method/analysis in combination with these WRF simulations.

- Given these above comments, I think part of the conclusion is already known a-priori by considering existing literature. One conclusion, that the change signals are much smaller than the bias should include how statistically significant the change signals are in relation to annual variability. Assume, for instance, that annual variability is larger than the change signal of mean precipitation. Would such a small change then be relevant to be discussed at all?

**Minor / Specific Comments**

- Some of the figures lack units on the y-axis (e.g. Fig. 5) and the legend is not correct (e.g. Fig.8/9), for instance "B/C reanalysis (ERA-I, R2)" suggests that ERA-Interim was bias corrected. Correct the legends and mention WRF, e.g. "B/C WRF reanalysis (ERA-I, R2)".
- Fig.2: (a) and (b) are exactly the same plots?
- Fig.2: add "GCM (ECHAM5,R1)" to the caption (d). And mention which configuration (R1-R3) was used for (b).
- Fig.3 & 4: add "WRF" to the figure caption, or to the y-axis label. Otherwise, it could be misread as GCM corrected output and not corrected WRF output.
- It would be clearer to label every subplot with (a), (b), and so on.
- Fig.8 & 9: I think the bottom row can be deleted. The changes in transition probability are already shown in the middle row. The percentage change of the probability change does not add to it.
- Fig.8 & 9: why show here WRF results from downscaling ERA-Interim, whereas in Fig.2 you show results from WRF downscaling NCEP?

- p.1, L.37: also mention statistical downscaling approach here?.
- p.2, L.5: delete "generates rainfall sequences by simulating physical climatic processes" and replace "does generate" by "generates".
- p.2, L.34: not only hindcast RCM rainfall, but also historical or in principle any data that is used for calibrating QQM.
- p.3, L.4: Teng et al. were not the first to use a double-gamma QQM, see for instance Yang et al. (2010, doi: 10.2166/nh.2010.004).
- p.3, L.12: do other studies agree with that? Otherwise, I suggest to add "for Australia".
- p.4, L.8-10: This sentence could be deleted as it is not of direct relevance to the study. If you want to keep this information, then move it to the end of the discussion section (or conclusion depending on what you hope to gain from it).
- p.5, L.37: Here and maybe on other locations you refer to "NARCliM rainfall" which is not correct, because NARCliM is no climate model. Better refer to e.g. "12-member WRF ensemble rainfall" or to WRF ("driving model", RX).
- p.6, L.9: I think you refer to Figure 5 here (not 6).
- p.6, L18: This also depends on the calibration period that was used (see main comments above).

- p.6, L.28ff: Is there any explanation for that and would it be sensitive if another BC method is used?
- p.6, L.31: how the WRF model/configuration was selected belongs to the data & methods section.
- p.6, L.34ff: Information on GR4J output belongs into the data & methods section.
- p.7, L.2-4: This information and discussion belongs into the Introduction (see main comments).
- p.7, L.6: add the information that "more bias" means in this case shorter wet spell durations. Then the next sentence is more meaningful.
- p.7, L.6: you might want to refer to Fig.5c here again.
- p.7, L.14: Do you refer to the middle row of Fig. 8 here?
- p.7, L.21: you mean WRF rainfall transition probabilities?
- p.7, L.21: somewhat unclear to what subplot of Fig. 9 you refer --> insert (a), (b), (c),... to the subplots?. From the sublots in the middle row I read a difference of about -0.02 to -0.04 for WRF (ERA-I,R2) and -0.04 to -0.1 for WRF (ECHAM5, R2). That is -2% to -4% and -4% to -10%. I do not see "over 10% over almost all of Victoria" from the plots. Or do you refer to the bottom right subplot? But, as I understand this figure, what is shown here is %-change of the difference. That means if the observed wet-wet prob. is e.g. 0.8, the difference in WRF is -0.08, then the %-change of this difference is -10%. But the error in wet-wet prob. is -8% and not -10%, isn't it? I think the bottom row plots are misleading, and I recommend to remove them. The information you want is already in the subplots of the middle row, isn't?
- p.7, L.41: I suggest to add this line to the legend in Fig.10 and also to explain in the caption that a position right to it means higher serial correlation.
- p.8, L.2f: a bias correction is not intended to correct rainfall occurrences (see my major comment).
- p.8, L.8f: Yes, you can compare the percentiles, as they are statistical measures of a distribution.
- p.8, L.16f: Is that a result of the seasonally applied BC? In what way is that problematic? Does not need to be a long discussion here, but briefly outline the problem associated to changed mean precipitation after BC. Would this change in mean be sensitive to a different BC method (e.g. monthly or annual)?
- p.8, L.24ff: are the found changes signals statistically significant in relation to annual variability?
- p.9, L.12-13: Does this affect the errors in the raw transition probabilities? (see major comment)
- p.9, L.14: unclear what is meant by "land and atmospheric circulation schemes". There is no atmospheric circulation scheme, you can either change the dynamical core or the physics, both of them then affect the circulation. Be more precise what is meant here.
- p.9, L16: First, the information that WRF R2 is the most credible configuration in terms of runoff modelling belongs to the data & methods section.
- p. 10, L. 26: "multi-".
- p. 10, L. 34f: again, BC cannot correct temporal errors.
- p. 10, L. 39: what is completely unmentioned is how statistically significant the change signals are in relation to annual variability.
- p. 10, L. 40: replace "dynamical downscaling process" with "RCM".

---

## Referee Report (RR2)

The authors analyse the effect of empirical bias correction on characteristics of daily precipitation from an GCM-RCM ensemble, which are relevant for runoff modelling. A seasonal empirical quantile-quantile mapping is applied to a historical and future period. A novel method for analysing the performance of bias correcting transition probabilities is presented. The bias correction performs as expected for rainfall amounts, but is not able to correct transition probabilities. Furthermore, the bias correction has only a minor effect on the climate change signals.

The paper has improved substantially, and I recommend to publish it after some minor issues are corrected.

**Minor issues**

1. On page 1 in line 37, the sentence suggests that there is already fine scale information in the GCM that just needs to be extracted. Revise this sentence to make clear that GCMs are too coarse to provide the fine scale details needed.

2. It is somehow unclear what observations were used to calibrate the QQM. I suppose it was AWAP, but mention it explicitly in section 2.3.

3. To me it is also occasionally unclear whether the NCEP-downscaled WRF data were used in the figures or only the GCM-downscaled WRF data. For instance in Fig. 5, where the caption only states "(b) absolute bias". State more precisely in the captions or figure titles which data went into the figure.

4. The calibration period used to construct the QQM is identical to the period it is applied to (1990–2009), page 5 in line 21 to 22. It is therefore clear, in particular as empirical QQM is used, that overfitting affects the results. This should be discussed in the Discussion section.

5. Remove redundant information in the text that describes the figure content. For instance, on page 6 line 8 to 9; page 8 on line 8 to 9; on page 7 in line 29 to 30. The figures should be self-explanatory.

6. On page 2 in line 13 to 14, the sentence is redundant to line 25 to 26. Just include Themeßl et al. to the listed references.

7. Add zero lines in Fig. 5 and Fig. 12 for better distinguishing positive and negative values.

8. On page 8, line 6 and 7, remove the last bullet point. The sentence makes no sense, because one reads "The bias in wet-wet transition probabilities is more problematic for modelling runoff [...] because [...] [it] increases the magnitude of the bias in wet-wet transition probabilities".

9. Add Haerter et al. (2011; doi: https://doi.org/10.5194/hess-15-1065-2011) as reference for the sentence on page 11 in line 3 to 4, who first showed that bias correction affects different time scales.

10. Some parts of the Conclusion should be moved to the Discussion section. For example, page 11 beginning in line 29.

11. "This" should always be followed by a subjective to which it refers. Otherwise, the sentence is imprecise and prone to ambiguity. Examples are on page 1 in line 15 and 21; page 2 in line 19 and 34; page 7 in line 26; page 10 in line 10; page 11 in line 31 and 36; page 12 in line 7).

    For instance page 7 in line 26:

    "[...]. This results in ...[]"

In this example, 'This' can either refer to 'low residual bias in dry-dry transition probabilities', or to 'more bias in wet-wet transition probabilities'. Small revisions would make such sentences more precise, e.g.:

"[...]. This reduced wet-wet transition probability results in ...[]"

12. Correct the punctuation where "however" is used in a compound sentence (e.g. page 1 in line 29 and 35; page 3 in line 19; page 10 in line 7) from

"[...], however ...[]"

to

"[...]; however, ...[]"

.

13. You might want to revise some long and bulky sentences to make them more concise and concrete, which improves readability. For instance on page 8 in line 32 to 35, but there are more cases.

---

## Editor Decision (ED1)

[revised manuscript text omitted]

**Figure 10: An alternative perspective on quantile-quantile mapping: daily rainfall amounts and associated probabilities plotted in** *d-w* **space. Quantile-quantile mapping bias correction (red) maps daily rainfall amounts from the raw data (blue) to the probability contours (dotted lines) corresponding to the appropriate observed daily amount (green). The dashed diagonal line represents** *p=d* **and hence an independent series of events (see sect. 2.4). Points lying above or to the right represent quantiles with greater (more persistent) autocorrelation.**

[Figure]

**Figure 11: Change signal (percentage difference of RCM future relative to RCM historical) in (a) rainfall percentiles; (b) mean annual, seasonal and monthly rainfall; (c) rainfall-sequencing metrics both before and after bias correction.**

---

## Author Response (AR2)

**Authors' Response and Marked-up Manuscript**

**Reviewer #2**

The revised version of the paper now includes some of the proposed changes from all reviewers. It has definitely improved, also in terms of reducing the number of figures, so that a clearer structure emerged.
5  Nevertheless, in my opinion there are still some open issues – in particular concerning the expectations on bias correction (BC) / hypothesis of the analysis, model/data description, and method description. These issues are either still unclear or insufficiently presented, but are directly relevant to the research question of the paper: what is the effect of BC on rainfall characteristics relevant for runoff modelling? Therefore, I still suggest that major revisions are required.

10          *Thank you for your insightful review and we have revised the manuscript accordingly as described in our response that follows.*

In the following, I will outline these issues.
**Major comments**
*   The Introduction still lacks some relevant aspects. Most of all, the authors should state what a BC is
15      meant to correct (amount of precipitation), what cannot be corrected (e.g. temporal errors, such as persistence of transition probabilities), and what is known to be affected by BC (change signals, persistence in some studies) and whether this effect is intended. Based on this literature review (which is still scattered over the whole text), hypotheses/expectations should be formulated that guide the analysis. For instance, it could be expected that a BC would remove the systematic errors in rainfall
20      amounts, but would not correct the transition probabilities or 3-day precipitation totals. Since other studies found that BC affects persistence and transition probabilities, this should be a motivation for the study and for introducing the new diagnostic presented by the authors.

          *We add "Most bias-correction methods alter daily amounts with the application of distributional mappings. The temporal structure of the occurrences is most often unaltered.*
25          *Further, bias correction can and does affect the magnitude of change signals (Hagemann et al., 2011; Gutjahr and Heinemann, 2013; Dosio, 2016)." Before "The underlying assumption…" on p.2, also adding "Given that QQM bias-correction approaches are specifically tailored for correcting daily amounts while retaining the existing sequencing of occurrences, we seek to understand the effect of bias correction on rainfall sequencing and*
30          *transition probabilities." in the final paragraph of the introduction.*

*   As another motivation, previous results with the here used WRF R1-R3 simulations should be summarized already in the Introduction. Obvious would be the results of Jin et al. (2016), who evaluated rainfall (and temperature) of the same WRF simulations on climate time scale. Jin et al. even concluded that a bias correction is required, which renders a natural motivation for this study. I further
35      noticed that the WRF simulations presented here actually stem from a double nesting (50km CORDEX Australasia domain and the inner 10km NARCliM domain, Fig.1 in Jin et al.). It would be good to write this in the model description – maybe also mention that it is the same simulations as in Jin et al.

          *We included Ji et al., 2016 in the discussion, but also including in the introductory sections is a good idea, this information has been included in section 2.1 and 2.2, which discusses the*
40          *NARCliM data.*

*Also, included "The NARCliM modelling domain is most of south-eastern Australia and the neighbouring Pacific Ocean at 10km × 10km resolution, nested within the larger CORDEX Australasian region (Giorgi et al., 2009) at 50km × 50km resolution (Evans et al, 2014)." in section 2.1*

*(Note most of the model-specific information is originally from Evans et al., 2014)*

- Other important evaluation results of the here used WRF simulations, such as presented on p. 9, L. 6ff of the Discussion, should be presented already in the Introduction or model description. Thereby, note that Gilmore et al (2016) only evaluated 15-day simulations focusing on a specific extreme precipitation event; thus, it is not guaranteed that WRF behaves similar at the daily scale for other synoptic
10    situations. These evaluations should then constitute the basis from where you start your analysis. Of particular importance is the information that WRF R2 renders the best configuration for hydrological applications (p.9, L.13-15) (why?) with respect to Olson et al. (2016). This information has to go into the Introduction, or into the model/data description. A natural question arises then whether your analysis support their conclusion? Finally, make clear how the models have been selected (e.g. as it is done in
15    the Introduction of Jin et al., 2016) and present this information in the model/data description. Only buried in the text (p.6, L.31), you state that WRF selection was based on credible simulations of 2-week heavy-precipitation periods. Do you then expect that these configurations faithfully represent other rainfall metrics, e.g. transition probabilities? If for instance transition probability/persistence of rainfall was not a selection criteria, but is important for hydrological applications, then the a-priori expectation
20    should be that BC would not help and a possible conclusion from your results could be e.g. that hydrologically relevant metrics should be added to a GCM/RCM selection process (based on CMIP5 etc.).

*We have moved the citations dealing with WRF performance in NARCliM to the introduction. We argue that the CMIP3/CMIP5 evaluation citations belong in the discussion as these are*
25    *general, regional results (i.e. not specific to NARCliM) relating to a previous reviewers' comment about suitability of climate models (GCMs and RCMs) to representing regional climate. NARCliM GCM/RCM evaluations directly relevant to NARCliM are already described/cited (Evans et al. papers) in the introduction.*

*We move the Olson et al. suggestion to use R2 in the introduction. Discussion on use of R2 for*
30    *runoff generation is included in Charles et al., in review, and we cite that here.*

*A general description of NARCliM model selection is provided in section 2.1, with relevant citations to Evans et al., and Ji et al. papers. We also include "WRF models were evaluated against 2-week heavy rainfall events with an intent to select the best possible RCM configurations for rainfall generation while also accounting for model uncertainty (Olson et*
35    *al., 2016)." in section 2.1, noting that the NARCliM modellers consider the selection procedure their "best attempt at reasonably simulating the climate of the south-east Australia while accounting for model uncertainty." (Olson et al., p. 224). As such, we don't a priori expect specific rainfall metrics to be properly modelled, but do suppose that NARCliM provides us with climatically faithful representations of SE Australia, hence the production of*
40    *this manuscript, i.e. to investigate secondary rainfall characteristics (transition probabilities, etc.) after the required bias correction.*

- In the discussion (p.8,L.28ff) you state that WRF has relatively large rainfall biases. You then conclude that there are model errors whose origin needs to be analysed and whether they render the physics implausible. In contrast, on p.8, L.37, you write these WRF simulations have been extensively tested,

and on p.9, L.6f that there is reasonable confidence in these simulations, although they have a general cold and wet bias. So, I guess the origin of these errors is understood to some degree and was already attributed in other studies?

*The rainfall amounts are acknowledged to be significantly biased, but the spatial occurrences, seasonal and interannual dynamics are recognised to be reasonably well modelled for Australia (as per the provided citations). The precise origin of the biases is suggested to include problems with subgrid cloud cover parameterisation (García-Díez et al., 2015; Di Virgilio et al., 2019), and is the subject of ongoing research. For our purposes we contend that the WRF modelling ensemble provides climatically and physically realistic representations of the study area's climate, albeit with a bias that needs to be corrected.*

*García-Díez, M., Fernández, J., and Vautard, R. An RCM multi-physics ensemble over Europe: multi-variable evaluation to avoid error compensation, Clim. Dynam., 45, 3141–3156, 2015, https://doi.org/10.1007/s00382-015-2529-x*

*Di Virgilio, G., Evans, J. P., Di Luca, A., Olson, R., Argüeso, D., Kala, J., Andrys, J., Hoffmann, P., Katzfey, J. J., Rockel, B., Evaluating reanalysis-driven CORDEX regional climate models over Australia: model performance and errors, Clim. Dynam., 53, 2985–3005, 2019, https://doi.org/10.1007/s00382-019-04672-w*

*We add "It is suggested that the wet bias (Fig. 3a) is related to subgrid cloud cover representations (see, e.g., García-Díez et al., 2015; Di Virgilio et al., 2019) and correction of this is the subject of current, ongoing research (Di Virgilio et al., 2019)." to the second paragraph of the discussion (p.9)*

- On p. 8, L.2f you raise the need for bias correction methods that correct the occurrences of rainfall events – in other words, the temporal structure. Once again, temporal errors indicate fundamental model errors that cannot be corrected by BC methods (e.g. Maraun et al., 2016; Maraun and Widmann, 2018). They are directly linked to errors in the dynamics of the climate model, which could result from many different sources, e.g. from coarse topography that affects the dynamics in a wrong way or missing relevant regional processes. BC is not meant to correct these errors. In contrast, on p.2, L.24f you state: "The underlying assumption of bias correction is that the RCM output faithfully represents climate processes relevant for rainfall, although the amounts themselves may not be accurate"; and on p.3, L. 15ff: "QQM bias correction cannot remove biases in rainfall sequencing …". This implicitly implies that the temporal aspect (persistence, transition probabilities, etc.) is assumed to be realistically simulated by the RCM and should thus be part of the model evaluation procedure prior to any BC. Any attempt to correct temporal errors of the model, however, could introduce unwanted major artefacts.

*We agree with these sentiments. In an ideal world the models would be faithfully reproducing the correlation structure of the rainfall fields. However, this appears to not necessarily be the case, and we seek to mitigate this. We approach the problem from a practical point of view, in terms of analysing and hopefully correcting data that is available for hydrological projections applications. Our research questions are: i) what climate data is available for hydrological applications? ii) what are the benefits and limitations of using this data? and iii) how to properly bias correct if necessary.*
*We are not aware of other studies that have analysed and evaluated temporal structure of rainfall (at least in the study area), and indeed all the bias correction and projection research we have reviewed have (implicitly) assumed that the correlations are correct. Our research findings will hopefully lead to better appreciation of the necessity of evaluating correlation*

*structure and/or methods (and their limitations) for dealing with this problem for future hydroclimate projections.*

- A motivation is missing why the authors repeat the bias correction with an empirical QQM, when there already is a corrected rainfall set using a double-gamma QQM (p.4, L.14f) that is provided by Evans and Argüeso (2014). Is there any reason not to use this data set? At least, provide a statement why this already existing data set was not used.

*In assessing the effect of bias correction, we wished to apply a controllable method in order to understand fully the method and its effects, rather than relying on previous methods. In this way we have better understanding of the way bias correction is applied and its flow-on effects. Since Teng et al. showed there is little difference between methods, we have no reason to believe that this has any impact on the findings of the paper.*

- The description of the models is still somewhat confusing. Please clearly refer to WRF simulations instead of either "bias-corrected reanalysis" or "bias-corrected GCM rainfall" (e.g. p.7, L.10f). Otherwise, it is very confusing (also in the figure labels), and it is just wrong because reanalysis/GCMs were downscaled with WRF and not directly bias corrected.

*This has been corrected in the line indicated and throughout, as well as the headings in figure 8 and 9.*

- I further recommend to revise section 2. First, all models that are used in the paper should be named here, including the runoff model GR4J that is first introduced in the results section (section 3.2 on p. 6, L.35). Then provide a short description of the WRF model with exact model version, model resolution, number of vertical levels, etc., and in particular briefly mention the differences of the WRF R1 – R3 configurations. It is still unclear to the reader whether R1 - R3 are different realisations based on slightly different initial fields or on different physics. Part of that information (e.g. model resolution, time period, etc.) is provided in section "2.2 Daily data", but that is not the correct place (maybe rename 2.2 to "Observations"). Furthermore, on p.4, L.6f, be more precise in describing that you use these three (R1-R3) WRF configurations for each of the four chosen GCMs, resulting in 12 simulations in total. One could misread p.4, L.6f that you downscale only three GCMs and that these downscaled WRF simulations are labelled R1-R3. Mention also that these result from double-nesting into the GCM/NCEP: 50km CORDEX-Australasia domain and 10km NARCliM domain. In section 2.2 only NCEP is stated as the reanalysis data used for this study, but in Fig. 8 & 9 you show results from WRF nested into ERA-Interim?

*As mentioned elsewhere, we prefer not to describe GR4J in too much detail given that it is not the main focus of the paper and is described elsewhere (citation given).*

*Increased detail of the WRF model version, resolution, and citation to specifics of R1-R3 physics configurations is provided in section 2.1*

*Included "The full NARCliM model ensemble thus consists of twelve members (four selected GCMs, each downscaled by the three RCM configurations)." in order to precisely indicate the size of the model ensemble.*

*Double-nesting of the modelling domains has been included (see response above).*

*Finally, ERA-I was also downscaled by the NARCliM project at a later date; as mentioned below, we have updated figures 8 and 9 using NCEP/NCAR reanalysis to be consistent with the other figure. Thus there is no need to include information on ERA-I in the present manuscript.*

- The description of the BC method still lacks some critical information. Although a reference to Teng et al. (2015) was added to the Introduction (but not to section 2.2?), it is still not clear how exactly the QQM was constructed for the here presented WRF simulations. Teng et al. (2015) use different time periods, different observations to construct the QQM, and different WRF simulations. Apart from the different distribution, is the QQM method identical to the QQM of Evans and Argüeso (2014), as suggested on p.4, L.12 ("... which is similar in many respects to the non-parametric procedure we apply ...")? Did you use the full historical period (1990-2009) to construct QQM and then applied it to the complete period? Or did you split it into calibration (dependent) and validation (independent data) subsets and then apply the so constructed QQM to the complete period? If you did the first without validating QQM with independent data, you cannot judge how strong overfitting is. If the complete historical period was used to construct the transfer functions, then a BC always reduces the mean error to zero (neglecting linear interpolation from 'qmap'), but the overfitting only takes effect on independent data (future time period data). Thus, overfitting could be large depending on how the QQM was constructed. Without such information, it is not possible to repeat the method/analysis in combination with these WRF simulations.

*Updated information is included in section 2.2 (citation to Teng) and section 2.3 (specifically that the full historical period was used to calibrate the transfer functions). We agree that with this setup it is not possible to judge overfitting, hence why bias in rainfall amount metrics (Figure 3 and 5) is effectively reduced to zero. We have added a parenthetical sentence acknowledging this when discussing figure 3 in section 3.2.*

*Even with zero calibration-validation error from applying BC to the same period, we see errors in transition probabilities (hence the motivation of this manuscript). Teng et al. looked specifically at the cross-validation error using two time different historical periods, and our manuscript effectively complements Teng's analysis by considering error/bias in secondary rainfall characteristics separately. We agree that in applying the WRF-BC method for end-user applications, proper calibration-validation methods should be applied to correctly analyse the error.*

- Given these above comments, I think part of the conclusion is already known a-priori by considering existing literature. One conclusion, that the change signals are much smaller than the bias should include how statistically significant the change signals are in relation to annual variability. Assume, for instance, that annual variability is larger than the change signal of mean precipitation. Would such a small change then be relevant to be discussed at all?

*We address this comment below.*

**Minor / Specific Comments**

- Some of the figures lack units on the y-axis (e.g. Fig. 5) and the legend is not correct (e.g. Fig.8/9), for instance "B/C reanalysis (ERA-I, R2)" suggests that ERA-Interim was bias corrected. Correct the legends and mention WRF, e.g. "B/C WRF reanalysis (ERA-I, R2)".

*y-axis in figure 5 is unitless as these are relative errors (percentages): include "as percentages" in caption to figure 5.*

*Titles on figures 8 and 9 have been updated as suggested*

- Fig.2: (a) and (b) are exactly the same plots?

*Thank you for noticing, this error crept in while preparing the figures for the revision, and has been redrawn as per the original version*

- Fig.2: add "GCM (ECHAM5,R1)" to the caption (d). And mention which configuration (R1-R3) was used for (b).

*Figure 2 has been updated as suggested.*

- Fig.3 & 4: add "WRF" to the figure caption, or to the y-axis label. Otherwise, it could be misread as GCM corrected output and not corrected WRF output.

*Captions updated*

- It would be clearer to label every subplot with (a), (b), and so on.

*Each subplot is not discussed separately and we prefer to keep the labelling as is due to the number of panels in figures 3 and 4.*

- Fig.8 & 9: I think the bottom row can be deleted. The changes in transition probability are already shown in the middle row. The percentage change of the probability change does not add to it.

*We agree, the bottom row has been deleted, and reanalysis has been updated to be consistent with figure 2.*

- Fig.8 & 9: why show here WRF results from downscaling ERA-Interim, whereas in Fig.2 you show results from WRF downscaling NCEP?

*Figures 8 and 9 have been updated to be consistent with figure 2, i.e. NCEP/NCAR reanalysis*

- p.1, L.37: also mention statistical downscaling approach here?.

  *A small sentence is included mentioning statistical downscaling*

- p.2, L.5: delete "generates rainfall sequences by simulating physical climatic processes" and replace "does generate" by "generates".

  *Done, thank you*

- p.2, L.34: not only hindcast RCM rainfall, but also historical or in principle any data that is used for calibrating QQM.

  *"Modelled historical rainfall (in this case" inserted before "hindcast RCM rainfall"*

- p.3, L.4: Teng et al. were not the first to use a double-gamma QQM, see for instance Yang et al. (2010, doi: 10.2166/nh.2010.004).

  *Citation added*

- p.3, L.12: do other studies agree with that? Otherwise, I suggest to add "for Australia".

  *Added ", at least in the Australian context studied by Teng…"*

- p.4, L.8-10: This sentence could be deleted as it is not of direct relevance to the study. If you want to keep this information, then move it to the end of the discussion section (or conclusion depending on what you hope to gain from it).

  *We believe the fact that ongoing research is providing better and updated products is still relevant to the underlying issues of bias correction and future projections, and as such we have moved the indicated sentence to the $2^{nd}$ paragraph of the discussion.*

- p.5, L.37: Here and maybe on other locations you refer to "NARCliM rainfall" which is not correct, because NARCliM is no climate model. Better refer to e.g. "12-member WRF ensemble rainfall" or to WRF ("driving model", RX).

  *We agree that NARCliM is not a climate model, however we argue that "NARCliM rainfall" is meaningful since it refers to an ensemble from the NARCliM project. However, we agree that it may lead to confusion so we include the following: "Daily accumulated precipitation*

*from the NARCliM 12-member WRF-downscaled ensemble (referred to from here onwards as "NARCliM rainfall") is produced…" in section 2.2.*

- p.6, L.9: I think you refer to Figure 5 here (not 6).

*Corrected, thank you*

5 - p.6, L18: This also depends on the calibration period that was used (see main comments above).

*We agree with this comment as noted in our response to the major comment above. We acknowledge this by adding "The quantile-quantile bias correction method is formulated to correct historical quantiles of rainfall exactly (when applied to the same time period used for calibration)…" in section 3.2.*

- p.6, L.28ff: Is there any explanation for that and would it be sensitive if another BC method is used?

*There is no particular explanation suggested here, and it would depend presumably on the correlation structure of the RCM rainfall. We suggest that it is unlikely to be different if another (daily) BC method is used, since the corrections applied from different daily BC methods is similar as noted in the introduction (see also Teng et al., 2015).*

- p.6, L.31: how the WRF model/configuration was selected belongs to the data & methods section.

*We agree, however the point mentioned in the manuscript is relevant in the results section as calibration to the largest historical rain events would tend to reduce the bias at very large (historical) rainfall accumulations*

- p.6, L.34ff: Information on GR4J output belongs into the data & methods section.

*We agree on this as well; runoff modelling is described in the companion paper that we cite, and so we reduce this to "runoff was modelled... using GR4J (see Charles et al., 2019 for more details)." Since runoff is not the key focus of the current manuscript, we decided not to describe the runoff modelling in greater detail here or in the introduction.*

- p.7, L.2-4: This information and discussion belongs into the Introduction (see main comments).

*Agreed, and this has been moved to the relevant paragraph in the introduction*

- p.7, L.6: add the information that "more bias" means in this case shorter wet spell durations. Then the next sentence is more meaningful.

*Added "(i.e. they are closer to zero)"*

- p.7, L.6: you might want to refer to Fig.5c here again.

*Cross reference added*

- p.7, L.14: Do you refer to the middle row of Fig. 8 here?

*Yes, and this reference has been included*

- p.7, L.21: you mean WRF rainfall transition probabilities?

- p.7, L.21: somewhat unclear to what subplot of Fig. 9 you refer --> insert (a), (b), (c),... to the subplots?. From the sublots in the middle row I read a difference of about -0.02 to -0.04 for WRF (ERA-I,R2) and -0.04 to -0.1 for WRF (ECHAM5, R2). That is -2% to -4% and -4% to -10%. I do not see "over 10% over almost all of Victoria" from the plots. Or do you refer to the bottom right subplot? But, as I understand this figure, what is shown here is %-change of the difference. That means if the observed wet-wet prob. is e.g. 0.8, the difference in WRF is -0.08, then the %-change of this difference is -10%. But the error in wet-wet prob. is -8% and not -10%, isn't it? I think the bottom row plots are misleading, and I recommend to remove them. The information you want is already in the subplots of the middle row, isn't?

  *We agree, and the bottom rows of fig. 8 and fig. 9 have been removed; on reflection percentage changes of probabilities are unclear and misleading.*

- p.7, L.41: I suggest to add this line to the legend in Fig.10 and also to explain in the caption that a position right to it means higher serial correlation.

  *Fig. 10 caption updated to include "The dashed diagonal line represents p=d and hence an independent series of events (see sect. 2.4). Points lying above or to the right represent quantiles with greater (more persistent) autocorrelation."*
  *The relevant sentence(s) on p.7 have been emended to reflect the revised caption.*

- p.8, L.2f: a bias correction is not intended to correct rainfall occurrences (see my major comment).

  *We definitely agree with the reviewer on this. However, bias correction is ultimately intended to provide physically plausible rainfall, of which rainfall occurrences are then indirectly assumed to be correct, which as we have demonstrated is not necessarily true. We alter "We surmise that, as bias correction is ultimately intended to produce physically plausible rainfall, bias correction methods…"*

- p.8, L.8f: Yes, you can compare the percentiles, as they are statistical measures of a distribution.

  *Yes, they can be compared directly, but we argue that it is not meaningful to do so since the underlying amount (which is the real quantity of interest) is different. "compared directly" changed to "meaningfully compared"*

- p.8, L.16f: Is that a result of the seasonally applied BC? In what way is that problematic? Does not need to be a long discussion here, but briefly outline the problem associated to changed mean precipitation after BC. Would this change in mean be sensitive to a different BC method (e.g. monthly or annual)?

  *The problem comes from deciding which is the "true" change signal in relevant quantities. Are raw change signals more realistic than bias corrected change signals? How can we tell?*

*A "small" change in future projections can in theory have large real-world consequences in terms of water planning and decision making. There is considerable debate on this in the literature, and we provide a cross reference to the discussion section where this is alluded to:*

*"applications (a discussion on bias correction effects on change signals is considered in the next section, and Charles et al. (2019) discuss this in the context of the present study in more detail)."*

- p.8, L.24ff: are the found changes signals statistically significant in relation to annual variability?

*Ordinarily we would report significance for any changes discussed in the manuscript. However, since the focus of this manuscript is change signals in relation to bias and bias correction, rather than in relation to annual variability, we prefer not to discuss this directly, however, Olson et al. (2016) analysed the statistical significance specifically, and we include a citation in section 3.4: "... SON rain which has a decrease projected by the NARCliM ensemble of around 20% (found to be statistically significant by Olson et al., 2016)."*

- p.9, L.12-13: Does this affect the errors in the raw transition probabilities? (see major comment)

*STR is a major determinant of the amount and timing of the seasonal rainfall progression through the year in SE Australia, and is important for correctly modelling climate in SE Australia. We do not expect that modelling this well will have any particular impact on transition probabilities.*

- p.9, L.14: unclear what is meant by "land and atmospheric circulation schemes". There is no atmospheric circulation scheme, you can either change the dynamical core or the physics, both of them then affect the circulation. Be more precise what is meant here.

*This section has been moved to the introduction in response to other comments, and "land and atmospheric circulation schemes" has been deleted. Since it is now mentioned directly in relation to the discussion on R1-R3 this should now be more clear.*

- p.9, L16: First, the information that WRF R2 is the most credible configuration in terms of runoff modelling belongs to the data & methods section.

*These sentences have been moved to section 2.1 and corrected*

- p. 10, L. 26: "multi-".

*Corrected to "multi-decadal"*

- p. 10, L. 34f: again, BC cannot correct temporal errors.

*We change: "biases in rainfall occurrences (such as rainfall autocorrelation, dry-dry and wet-wet transition probabilities) are retained and in some cases increased are not properly corrected with QQM.*

- p. 10, L. 39: what is completely unmentioned is how statistically significant the change signals are in relation to annual variability.

*Kindly see our response to the similar comment above.*

- p. 10, L. 40: replace "dynamical downscaling process" with "RCM".

*Replaced*

**Reviewer #3**

The authors missed the point regarding bias with respect to spatial correlation or persistence. They discuss the "inflation" issue as raised by Maruan (2013). This is regarding the inflation of the subgrid aggregated precipitation amount. This stems from applying the same quantity from a lower-resolution RCM/GCM to all of the points or grids that fall within each low-res grid cell. So, for example, when the model simulates 100 mm for a day, this amount is similarly bias corrected for all subgrid cells contained within the larger model grid cell. This often will lead to inflating the area-average, because the subgrid cell variability                    is                    not                    well                    represented.

Entirely separate, but equally important, is the issue of modeled versus observed spatial correlation (or what the authors refer to as spatial structure) when they are on the same grid (Bardossy and Pegram, 2012). Consider an example where both the model and observations are on the same 10x10 grid. Also, consider the model to have, on average, larger spatial correlation (correlation in time between each grid cell, this would be a 100x100 correlation matrix) than observations (i.e., when it rains at one grid cell, it is much more likely to rain at an adjacent grid cell than what observations show). This effect will translate to inflating the area-averaged precipitation amounts of the bias-corrected modeled data. This is because the model has larger covariance or spatial correlation or coherence or spatial structure than the observations, so the extent of individual model storms are, on average, greater than those seen in observations. The impact of the model over- or under-estimating the spatial correlation matrix, with respect the observations, is not effectively dealt with by bias correction. And again, this is an entirely different issue than the "inflation" issue discussed by Maraun (2013). The authors end the paragraph that begins page 10 with, "However, the issue of using a bias correction methodology that corrects daily amounts (and more generally temporal structure) while preserving spatial structure across catchments and basins remains a challenge and is a direction for further research." I would agree that a multivariate method that effectively deals with temporal and spatial biases simultaneously is a direction for further research. However, a researcher can correct for spatial structure via recorrelation (Bardossy and Pegram, 2012) and temporal structure via a commonly used univariate bias correction method. The result of correcting the biases of time and space independently will not perfectly correct all possible types of bias within the modeled data, but it will be superior to doing one or the other alone. At the very least, the impact that model bias (in spatial structure) has on area-averaged precipitation needs to be further acknowledged and discussed.

*We thank the reviewer for their comments, and acknowledge that we didn't treat the issue of underestimation of modelled spatial correlation sufficiently in the previous version. We add the following section to the discussion:*

[revised manuscript text omitted]

**Figure 10: An alternative perspective on quantile-quantile mapping: daily rainfall amounts and associated probabilities plotted in** *d-w* **space. Quantile-quantile mapping bias correction (red) maps daily rainfall amounts from the raw data (blue) to the probability contours (dotted lines) corresponding to the appropriate observed daily amount (green). The dashed diagonal line represents** *p=d* **and hence an independent series of events (see sect. 2.4). Points lying above or to the right represent quantiles with greater (more persistent) autocorrelation.**

[Figure]

**Figure 11: Change signal (percentage difference of RCM future relative to RCM historical) in (a) rainfall percentiles; (b) mean annual, seasonal and monthly rainfall; (c) rainfall-sequencing metrics both before and after bias correction.**

---

## Author Response (AR3)

**Authors' response to review**

**Response to editor**

The title should be more specific to the main core of the paper. At least the fact that dynamical downscaling is explored should be mentioned. Something in the direction of: Bias in dynamic downscaled ...

5 *We have altered the title to "Bias in dynamically downscaled rainfall characteristics for hydroclimatic projections"*

"as well as"

*We have revised second sentence of the introduction and have corrected uses of "as well as" throughout.*

10 The conclusion section is too long - please try summarizing the key take home messages in one or two paragraphs. The first paragraph, for example, is not need as it repeats the written in the introduction. Some other paragraphs should be embedded into the discussion section.

*The conclusion has been shortened, including: removing the redundant information in the first paragraph, moving paragraph 3 to the discussion section, shortening final paragraph.*

15 **Review of the paper "Bias in downscaled rainfall characteristics" by Potter et al.**

The authors analyse the effect of empirical bias correction on characteristics of daily precipitation from an GCM-RCM ensemble, which are relevant for runoff modelling. A seasonal empirical quantile-quantile mapping is applied to a historical and future period. A novel method for analysing the performance of bias correcting transition probabilities is presented. The bias correction performs as expected for rainfall amounts, but is not able to correct transition probabilities. Furthermore,
20 the bias correction has only a minor effect on the climate change signals.
The paper has improved substantially, and I recommend to publish it after some minor issues are corrected.

*We thank the reviewer for their review.*

**Minor issues**

1. On page 1 in line 37, the sentence suggests that there is already fine scale information in the GCM that just
25 needs to be extracted. Revise this sentence to make clear that GCMs are too coarse to provide the fine scale details needed.

*Sentence adjusted to read "Downscaling is the process by which finer scale spatial detail is produced  from the  GCM change information, which is at too coarse a resolution to be usable"*

2. It is somehow unclear what observations were used to calibrate the QQM. I suppose it was AWAP, but mention it explicitly in section 2.3.

*AWAP is stated as the observational data here*

3. To me it is also occasionally unclear whether the NCEP-downscaled WRF data were used in the figures or only the GCM-downscaled WRF data. For instance in Fig. 5, where the caption only states "(b) absolute bias". State more precisely in the captions or figure titles which data went into the figure.

*We assume this refers to fig. 7, as fig. 5 caption contains "NARCliM 12-model ensemble" in the description. (Please advise if this is not the case.) Figure 7 caption has been updated "Figure 7: Ensemble (downscaled GCM hindcasts from the NARCliM 12-model ensemble) median modelled runoff…"*

4. The calibration period used to construct the QQM is identical to the period it is applied to (1990–2009), page 5 in line 21 to 22. It is therefore clear, in particular as empirical QQM is used, that overfitting affects the results. This should be discussed in the Discussion section.

*We acknowledge this is true and have added two sentences (second and third sentences in the first paragraph in the discussion).*

5. Remove redundant information in the text that describes the figure content. For instance, on page 6 line 8 to 9; page 8 on line 8 to 9; on page 7 in line 29 to 30. The figures should be self-explanatory.

*Changes made*

6. On page 2 in line 13 to 14, the sentence is redundant to line 25 to 26. Just include Themeßl et al. to the listed references.

*Sentence adjusted as suggested*

7. Add zero lines in Fig. 5 and Fig. 12 for better distinguishing positive and negative values.

*Zero lines have been added in these figures*

8. On page 8, line 6 and 7, remove the last bullet point. The sentence makes no sense, because one reads "The bias in wet-wet transition probabilities is more problematic for modelling runoff [...] because [...] [it] increases the magnitude of the bias in wet-wet transition probabilities".

*Deleted, and sentence adjusted*

9.  Add Haerter et al. (2011; doi: https://doi.org/10.5194/hess-15-1065-2011) as reference for the sentence on page 11 in line 3 to 4, who first showed that bias correction affects different time scales.
* * *
*Citation added*
* * *
10. Some parts of the Conclusion should be moved to the Discussion section. For example, page 11 beginning in line 29.
* * *
*(sub-)Paragraph moved to first paragraph in discussion as suggested*
* * *
11. "This" should always be followed by a subjective to which it refers. Otherwise, the sentence is imprecise and prone to ambiguity. Examples are on page 1 in line 15 and 21; page 2 in line 19 and 34; page 7 in line 26; page 10 in line 10; page 11 in line 31 and 36; page 12 in line 7).
    For instance page 7 in line 26:
        "[...]. This results in ...[]"
    In this example, 'This' can either refer to 'low residual bias in dry-dry transition probabilities', or to 'more bias in wet-wet transition probabilities'. Small revisions would make such sentences more precise, e.g.:
        "[...]. This reduced wet-wet transition probability results in ...[]"
* * *
*We have gone through the manuscript and clarified instances of ambiguous "this".*
* * *
12. Correct the punctuation where "however" is used in a compound sentence (e.g. page 1 in line 29 and 35; page 3 in line 19; page 10 in line 7) from
        "[...], however ...[]"
    to
        "[...]; however, ...[]"
* * *
*Punctuation of instances of "however" in compound sentences have been changed as suggested*
* * *
13. You might want to revise some long and bulky sentences to make them more concise and concrete, which improves readability. For instance on page 8 in line 32 to 35, but there are more cases.
* * *
*This sentence has been adjusted, as have other sentences, after re-reading the manuscript.*
* * *

[revised manuscript text omitted]

**Figure 10: An alternative perspective on quantile-quantile mapping: daily rainfall amounts and associated probabilities plotted in**
**$d$-$w$ space. Quantile-quantile mapping bias correction (red) maps daily rainfall amounts from the raw data (blue) to the probability**
**contours (dotted lines) corresponding to the appropriate observed daily amount (green). The dashed diagonal line represents $p=d$**
**and hence an independent series of events (see sect. 2.4). Points lying above or to the right represent quantiles with greater (more**
**persistent) autocorrelation.**

[Figure]

**Figure 11: Change signal (percentage difference of RCM future relative to RCM historical) in (a) rainfall percentiles; (b) mean annual, seasonal and monthly rainfall; (c) rainfall-sequencing metrics both before and after bias correction.**

---

## Author Response (AR4)

Figure 1. Please make sure the labels in the plot are not covering the lines and symbols.

*Done*

Figure 2. Please add a scale-bar (in km) to one of the subplots.

*Scale bar added to panel (e)*

Figure 3. Please add a title and units to the color-scale (here, and in the other figures)

*Figure 3,4 colour scale title included, also figures 8, 9.*

Figure 5. The legend is not in a readable form, please correct.

*Figure redrawn*

Figure 8. I suggest withdrawing some of the labels on the scale. In addition, can it be placed to the right of the figure? Same
in Fig. 9.

*Done*

Figure 10. The plot will be clearer if the units ("mm") will be removed and the labels of each line will not cover each other.

*Done*

Figure 11. The legend is unclear.

*Figure redrawn*

[revised manuscript text omitted]

**Figure 10: An alternative perspective on quantile-quantile mapping: daily rainfall amounts and associated probabilities plotted in**
*d-w* **space. Quantile-quantile mapping bias correction (red) maps daily rainfall amounts from the raw data (blue) to the probability**
**contours (dotted lines) corresponding to the appropriate observed daily amount (green). The dashed diagonal line represents** *p=d*
**and hence an independent series of events (see sect. 2.4). Points lying above or to the right represent quantiles with greater (more**
**persistent) autocorrelation.**

[Figure]

**Figure 11: Change signal (percentage difference of RCM future relative to RCM historical) in (a) rainfall percentiles; (b) mean annual, seasonal and monthly rainfall; (c) rainfall-sequencing metrics both before and after bias correction.**